# Medical education and distrust modulate the response of insular-cingulate network and ventral striatum in pain diagnosis

Giada Dirupo[1,2,3]*, Sabrina Totaro[1], Jeanne Richard[3,4], Corrado Corradi-Dell'Acqua[1,2]

[1]Theory of Pain Laboratory, Department of Psychology, Faculty of Psychology and Educational Sciences (FPSE), University of Geneva, Geneva, Switzerland; [2]Geneva Neuroscience Center, University of Geneva, Geneva, Switzerland; [3]Swiss Center for Affective Sciences, University of Geneva, Geneva, Switzerland; [4]Department of Psychology, Swiss Distance University Institute, Brig, Switzerland

**Abstract** Healthcare providers often underestimate patients' pain, sometimes even when aware of their reports. This could be the effect of experience reducing sensitivity to others pain, or distrust toward patients' self-evaluations. Across multiple experiments (375 participants), we tested whether senior medical students differed from younger colleagues and lay controls in the way they assess people's pain and take into consideration their feedback. We found that medical training affected the sensitivity to pain faces, an effect shown by the lower ratings and highlighted by a decrease in neural response of the insula and cingulate cortex. Instead, distrust toward the expressions' authenticity affected the processing of feedbacks, by decreasing activity in the ventral striatum whenever patients' self-reports matched participants' evaluations, and by promoting strong reliance on the opinion of other doctors. Overall, our study underscores the multiple processes which might influence the evaluation of others' pain at the early stages of medical career.

*For correspondence: giada.dirupo@gmail.com

Competing interests: The authors declare that no competing interests exist.

## Introduction

Unrelieved pain is a major medical problem worldwide, resulting in human suffering and economic costs. Unlike other medical conditions, which are diagnosed through reliable tests, pain is difficult to quantify objectively, and it is mainly assessed by medical practitioners (physicians, nurses, medical students, etc.) using indirect information or self-reports. As such, healthcare providers systematically underestimate patients' pain (*Davoudi et al., 2008*; *Duignan and Dunn, 2008*; *Kappesser et al., 2006*; *Puntillo et al., 2003*; *Teske et al., 1983*), a phenomenon which emerges as early as during university (*Xie et al., 2018*), becomes more pronounced with long-lasting experience in the field (*Choinière et al., 1990*; *Davoudi et al., 2008*), and affects prevalently women (*Greenwood et al., 2018*) and ethnical minorities (*Ghoshal et al., 2020*; *Kaseweter et al., 2012*; *Todd et al., 2000*).

Despite extensive research, the causes of pain underestimation are still unclear. A popular theory suggests a major role of medical experience. Indeed, due to their daily exposure to the severe conditions, often characterized by high levels of suffering, healthcare providers could have progressively changed their frame of reference of what characterizes an extreme pain (*Bergh and Sjöström, 1999*). Hence, they might be inclined to provide less intense evaluations than those individuals (e.g. the average patient) who do not share the same experience. Within this framework, neuroimaging literature repeatedly implicated brain regions, such as the anterior insula (AI) and dorsal anterior cingulate cortex (dACC) in processing other people's pain (*Ding et al., 2020*; *Fan et al., 2011*; *Jauniaux et al., 2019*; *Kogler et al., 2020*; *Lamm et al., 2011*; *Timmers et al., 2018*). As this

network is partly common to that involved in first-hand pain (*Corradi-Dell'Acqua et al., 2016*; *Corradi-Dell'Acqua et al., 2011*; *Kogler et al., 2020*; *Lamm et al., 2011*; *Zhou et al., 2020*), scholars interpreted this activations in terms of empathy, whereby individuals simulate the observed state on oneself (*Bernhardt and Singer, 2012*; *Lamm et al., 2019*; *Stietz et al., 2019*). Critically, expert medical practitioners exhibited lower neural response in these regions (*Cheng et al., 2017*; *Cheng et al., 2007*), whereas they appear to over-recruit prefrontal structures, often implicated in regulation and control (*Cheng et al., 2007*), possibly reflecting enhanced ability at regulating their primary empathetic responses to the sight of people potential sufferance.

Medical experience and emotion regulation might not be sufficient for explaining the entirety of the phenomenon. In clinical settings, acknowledging high levels of pain often leads to the prescription of strong analgesics, which however have contraindications for patients' health (*Buckeridge et al., 2010*; *Butler et al., 2016*; *Makris et al., 2014*). Concerns for such side-effects (e.g. opiophobia) contribute to inadequate pain treatment in medical settings (*Bennett and Carr, 2002*; *Bertrand et al., 2021*; *Corradi-Dell'Acqua et al., 2019*), as healthcare providers prioritize those cases in which pain is unequivocally established. In this perspective, one study showed that doctors and nurses tended to underestimate pain in larger extent when presented with cues that patients might have lied or exaggerate their ratings (*Kappesser et al., 2006*). In a similar vein, recent qualitative investigations revealed that emergency nurses tend to privilege their own personal judgments rather than patients pain-ratings (*Johannessen, 2019*; *Vuille et al., 2018*). Furthermore, when their clinical assessment is inconsistent with the self-reports (e.g. patients denounce high suffering while few minutes before they were chatting/laughing), nurses try to convince patients to reappraise their pain, or at least mark their own point of view in the clinical chart when transferring the case to a colleague (*Vuille et al., 2018*). Thus, healthcare providers appear to scrutinize the reliability of patients' pain and, in some case, even to mistrust their reports and feedbacks, at the advantage of the opinion of other medical practitioners. However, to the best of our knowledge, no study investigated systematically the role played by patients' feedbacks in pain assessments, and whether this influence changes with respect to that of other physicians.

Psychology and neuroscience literature provided us with several tools to investigate the role of social influence in individuals' perceptual decisions and subjective experiences (*Cialdini and Goldstein, 2004*; *Klucharev et al., 2009*; *Schnuerch and Gibbons, 2014*; *Wu et al., 2016*). In a seminal paradigm, participants evaluated the attractiveness of women and, subsequently, saw a feedback representative of peers' mean ratings (*Klucharev et al., 2009*). Critically, when unexpectedly requested to repeat the task, participants reappraised each face consistently with the feedback received. Furthermore, a large discrepancy with others' evaluations increased the neural activity in the dorsal-posterior portion of the medial prefrontal cortex (dpMPFC), often implicated in prediction error, whereas agreement with these feedbacks recruited the reward system in the ventral striatum (VS) (*Klucharev et al., 2009*, see also *Wu et al., 2016* as meta-analysis). These effects have supported the idea that, even in subjective decisions with no correct/incorrect answers, participants considered the opinion of peers as an indirect proxy of their proficiency in the task (*Klucharev et al., 2009*).

The present research aims at testing the degree with which healthcare providers take into account feedbacks about their pain assessment, and whether they weight in different extent the opinion of the patient from that of other physicians. Across two-experiments, we implemented a modified version of the social influence paradigm described above (*Klucharev et al., 2009*; *Wu et al., 2016*). Here, individuals appraised the pain of facial expression video-clips, and subsequently were confronted with two feedbacks (see *Figure 1*): the self-report of the person in pain (the *Target* of each video) and the average opinion of 20 medical practitioners (hereafter, *MPs*). The two feedbacks occurred simultaneously, and each of them could appear either at a lower, equal or higher position with respect to participants' initial rating (see Materials and methods). After 30 min from the end of this first session, participants were unexpectedly asked to evaluate again the same facial expressions, thus providing the opportunity to assess if their initial evaluations changed and, in that case, which feedback (*Target* vs. *MPs*) best explained the reappraisal. In the context of pain assessments, the rating of the patient should represent the most reliable source of information to which compare one's performance. Hence, individuals' behavioral and neural responses should ideally be influenced in privileged fashion by the *Target*'s feedback, as opposed to that of *MPs* who are not experiencing the pain directly. However, healthcare providers might mistrust information arising

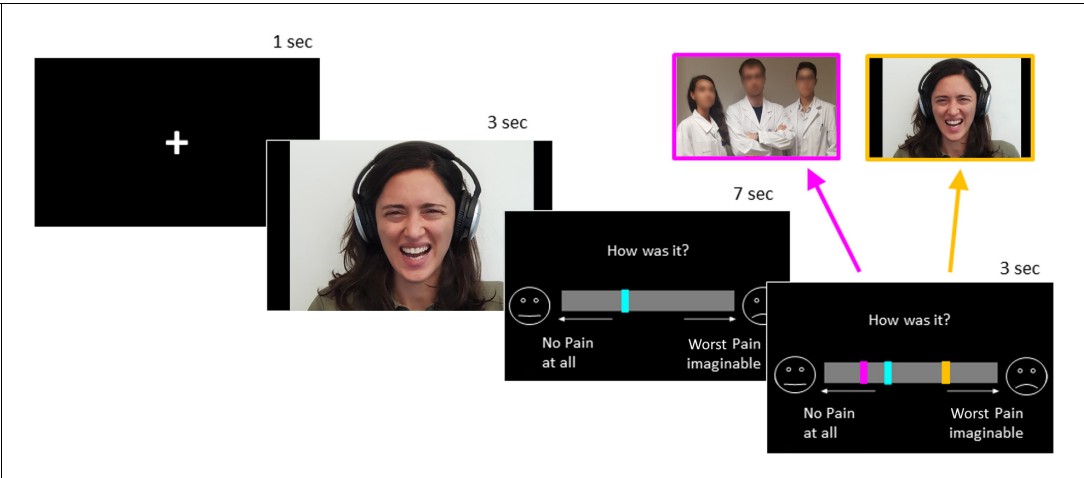

**Figure 1.** Schematic representation of trial structure in the first session. For demonstrative purposes, one author of the study (GD) is depicted in the figure expressing pain in similar way to the video-clips used as stimuli (*Lamm et al., 2007*).

from the person in pain, and privilege the one of physicians, toward which they have developed positive dispositions. Hence, when confronted with feedbacks about their pain assessment, they might value in greater extent the opinion of other doctors at the expense of that of the patients.

## Results

### Population validation

In the present study, we recruited as a population of interest medical students from different years of medical faculty. This population was chosen due to its feasibility in terms of recruitment, and the straightforward quantification of individual experience (corresponding to the year of enrollment at university, regardless of future specializations, department, hierarchy, etc.). Furthermore, pain underestimation has been documented also in medical students (at least after 3 years, *Xie et al., 2018*), thus making this population suitable for our research question. Finally, we assumed that these individuals would display stronger positive dispositions toward the category of physicians, relatively to lay individuals. This assumption was validated by a pilot experiment (Pilot 1, see Appendix 1) run on an independent population of 155 participants, organized in four groups: 38 Experienced Medical Students (EMS, from the 5th and 6th year), 32 Intermediate Medical Students (IMS, from the 3rd and 4th year), 56 Young Medical Students (YMS, from the 1st and 2nd year), and 29 lay Controls (from different faculties/professions, except those related to medicine, infirmary, dentistry and physiotherapy). Through an implicit association task (IAT), we calculated an index (D-score) reflecting strength of positive disposition toward the category of *MPs* (relative to non-*MPs*). An Analysis of Variance (ANOVA) aimed at testing for group differences revealed higher D-scores in EMS, relative to Controls and less experienced students (see *Figure 2A*; see also *Figure 2D* for a similar effect in subjects from Experiment 2).

### Pain expressions ratings

Having established that experienced medical students (EMS) are especially positively disposed toward the category of physicians (relative to other categories), we then tested whether they differed in the way in which they assess pain, and take into considerations feedbacks about their assessment. In Experiment 1, we recruited 120 participants, organized as 30 Controls, 30 YMS, 30 IMS, and 30 EMS. *Appendix 1—table 1* provides full details about demographic information of these participants, with controls displaying comparable age to EMS. As first step, we tested whether the mean pain rating prior to the presentation of any feedback changed as function of *Group* through an ANOVA with *Gender* and *Age* as nuisance control variables. The analysis confirmed an effect of *Group* ($F_{(3,114)}$ = 3.09; p=0.030). Post-hoc t-tests showed that YMS were associated with

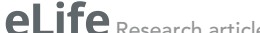

**Figure 2.** Behavioral Results of IAT and first session. Boxplots and individual data describing (**A–D**) the IAT D-score across groups (higher values refer to stronger implicit positive dispositions toward the category of MPs; see Appendix 1); (**B–E**), Pain Intensity Ratings from first session of the task; (**C–F**) the Distrust about pain authenticity from the final debrief session. In all plots, data are divided across groups, referring to Controls [C], Young Medical Students [YMS], Intermediate Medical Students [IMS] and Experienced Medical Students [EMS]. For each boxplot graph, the horizontal line represents the median value of the distribution, the star represents the average, the box edges refer to the inter-quartile range, and the whiskers to the data range within 1.5 of the inter-quartile range. Individual data-points are also displayed as dots. '**' and '*' refer to independent sample t-tests associated with p<0.01 and p<0.05, respectively.

higher ratings than both Controls ($t_{(58)}$ = 2.55; p=0.013) and EMS ($t_{(58)}$ = 2.79; p=0.007, see *Figure 2B*). No significant difference was found when testing all the other possible combinations of groups ($|t_{(58)}|$<1.75, p>0.082). We also found an effect of *Gender* ($F_{(1,114)}$ = 5.58; p=0.020), reflecting more pronounced ratings in female, relative to male, subjects (*Age:* $F_{(1,114)}$ = 1.27; p=0.271).

In Experiment 2, we repeated the same paradigm (with minor adjustments, see methods) by focusing specifically on 52 participants (26 Controls and 26 EMS), and by recording neural responses (through functional Magnetic Resonance Imaging [fMRI]). The pain ratings from this experiment did

not change as function of *Group* ($F_{(1,48)}$ = 0.84; p=0.369; all other effects $F$ < 1.36; p>0.249, *Figure 2E*), consistently with Experiment one when the same groups were compared (*Figure 2B*). We then looked at the neural activity and searched for regions sensitive to the 'painfulness' of the expressions (and not for the mere presentation of a face). This was achieved by testing effects significantly associated with the parametrical modulation of pain ratings (whilst controlling for *Gender* and *Age*). We first tested the main effect (activations common to both groups) and found a positive linear relationship at the level of amygdala extending to the periaqueductal grey (PAG), and to the fusiform gyrus (*Figure 3*, green blobs). Furthermore, when applying small volume correction in regions previously implicated in paradigms for pain empathy (*Kogler et al., 2020*, see methods) the left anterior insula (AI) was also found (see *Appendix 1—table 3* for full details). These effects were consistent with previous literature on emotional facial expressions (*Pessoa and Adolphs, 2010*; *Vuilleumier et al., 2004*; *Vuilleumier et al., 2001*; *Vuilleumier and Pourtois, 2007*), and on processing others' pain through the same networks of first-hand pain experience (*Corradi-Dell'Acqua et al., 2016*; *Corradi-Dell'Acqua et al., 2011*; *Kogler et al., 2020*; *Lamm et al., 2011*; *Zhou et al., 2020*). Subsequently, we analyzed group differences. We found stronger modulations in Controls as opposed to EMS in the dorsal portion of the anterior cingulate cortex (dACC). Under small volume correction, we also implicated the left ventral AI (*Figure 3*, red blobs). Overall, our data converge with, but also extend, previous evidence by showing that medical education leads to decreased sensitivity to others' pain (*Cheng et al., 2017*; *Cheng et al., 2007*; *Choinière et al., 1990*; *Davoudi et al., 2008*), both in terms of behavioral ratings (when comparing EMS to YMS) and in the neural response of predefined regions such as AI and dACC (when comparing EMS to Controls).

Up to now, we investigated inter-individual differences only in terms of group. However, it is possible that participants' performance might be better explained in terms of positive dispositions toward *MPs* (as measured only in Experiment two through the same IAT implemented in Pilot 1), in terms of individual *Distrust* towards the expressions' authenticity (measured in the debrief in Experiments 1 and 2), or in terms of individual empathic traits (measured through the Interpersonal Reactivity Index questionnaire; *Davis, 1980*). We repeated all the analyses carried out above, by replacing the *Group* factor with either the IAT D-Score, the *Distrust* estimate, or each empathic score as continuous regressors. Behavioral results were not systematic across experiments, with

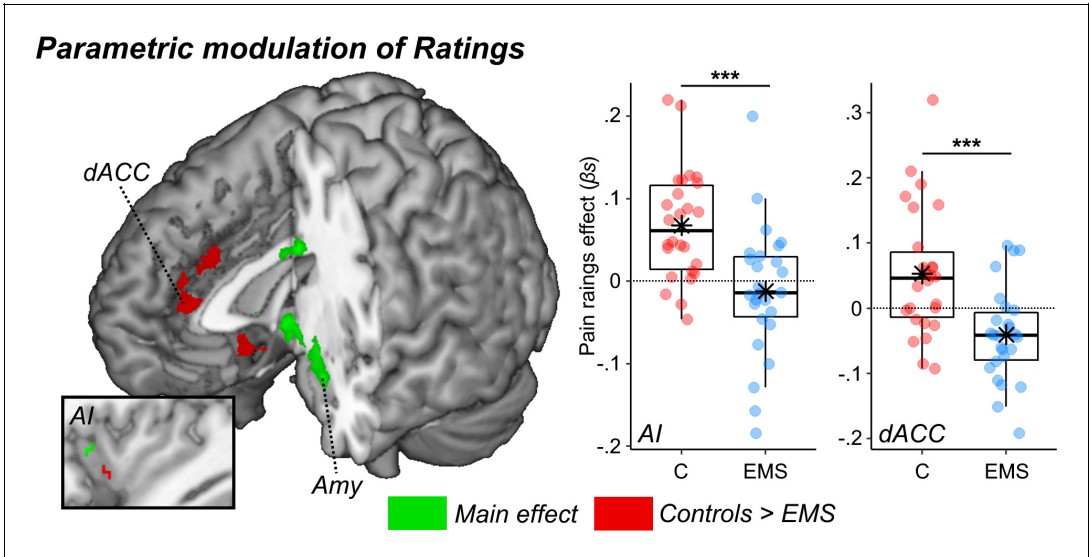

**Figure 3.** Surface renderings displaying regions implicated in the parametrical modulation of pain ratings. Effects are displayed under a height threshold corresponding to p<0.001, with each region surviving cluster-correction for multiple comparisons for the whole brain, or associated with a peak surviving small volume correction for a mask of interest (this is the case of the two insular activations). Green blobs describe regions observed when taking into account both groups (main effect), whereas red regions display stronger effects for Controls as opposed to EMS. Parameter estimates from two regions implicated in group differences are displayed through boxplots. AI: Anterior Insula. dACC: dorsal Anterior Cingulate Cortex; Amy: Amygdala.

*Distrust* influencing only Experiment 1 (Exp 1: $b = -0.31$, $F_{(1,113)} = 5.50$, p=0.021; $b = 0.02$, $F_{(1,44)} = 0.02$, p=0.884), and perspective rating scores only Experiment 2 (Exp 1: $b = 0.02$, $F_{(1,116)} = 0.60$, p=0.442; Exp 2: $b = 0.07$, $F_{(1,46)} = 4.06$, p=0.0497; all other covariates $F < 2.38$, p>0.123). We found no effect on the neural responses.

## Effect of feedback

We assessed whether participants evaluations were influenced by the presentation of the social feedbacks. For this purpose, we calculated a measure of Reappraisal, defined as the change in rating observed when participants were (unexpectedly) asked to evaluate the same facial expression again at the end of the experiment (see Materials and methods). The closer this value is to zero, the more participants' evaluation of a given face was stable across time. The more the value differed from zero, the more participants adjusted their evaluation.

We therefore run a multi-level regression to see whether single-trial Reappraisal changed as function of the relative position of the two social feedbacks, with the purpose of testing whether participants adjusted more strongly toward one feedback (*Target* vs. *MPs*), and whether such adjustment changed as function of *Group*. In both experiments, the analysis of Reappraisal revealed a main effect of both *Target* (Exp 1: $b = 0.13$, $F_{(1,67.99)} = 52.63$, $p < 0.001$; Exp 2: $b = 0.07$, $F_{(1,32.43)} = 5.70$, $p = 0.022$) and *MPs* (Exp 1: $b = 0.12$, $F_{(1,116.87)} = 19.40$, $p < 0.001$; Exp 2: $b = 0.11$, $F_{(1,53.92)} = 11.57$, $p < 0.001$) feedbacks. *Figure 4* shows Reappraisal values across three relative feedback positions on the scale. When each feedback was higher than participants' initial evaluation, a positive adjustment was observed. This was instead not the case when the feedbacks appeared in the same or lower positions of the scale (at least not systematically across experiments). No significance was associated with *Group, Gender* or *Age* ($F \leq 2.57$, p $\geq$ 0.053).

We then looked at the brain activity (from Experiment 2) evoked by the presentation of the feedbacks (see *Appendix 1—tables 4–6* for full details). Brain activity was tested under a similar rationale than Reappraisal, with the exception that in this case we modeled parametrically the absolute discrepancy between participants' ratings and the feedback, with the purpose of identifying regions sensitive to 'errors' in either direction. Within this framework, discrepancy effects were found in a widespread network involving dpMPFC and right AI extending to inferior frontal gyrus (IFG; see *Figure 5A*, white overlapping blobs). Importantly, this network was highly reminiscent with previous studies testing deviation effects of social influence on subjective evaluations (*Klucharev et al., 2009*; *Wu et al., 2016*) (see *Appendix 1—figure 1*), and appeared recruited for both kinds of feedback (*Target, MPs*). Visual inspection of the parameters extracted from dpMPFC (*Figure 5D*) suggest that dpMPFC could be slightly more sensitive to deviations toward the lower levels of the scale, rather deviations per se. We tested this hypothesis by running a follow-up model where signed (rather than absolute) discrepancy was specified instead. This allowed us to search for regions responding more strongly to positive than negative deviations of the same magnitude, and *vice versa*. We found no effect, neither for the Target feedback nor for the *MPs*. Hence, the modulations described in *Figure 5A* are better explainable in terms of absolute discrepancy.

We also investigated regions responding to the agreement between participants' own rating and the position of the feedback (as in previous studies on the same paradigm, *Klucharev et al., 2009*; *Wu et al., 2016*), by testing for negative modulations of the absolute discrepancy with the feedbacks: that is, areas responding the most when the gap with the social feedbacks was 0. For both the *Target* and the *MPs*, we found a widespread network which converged around the bilateral parietal and ventral occipital cortex (see *Figure 5B*, white overlapping blobs). Critically, for the *Target* feedback only, the ventral striatum (VS) and the Putamen were also found in line with previous observations (*Klucharev et al., 2009*; *Wu et al., 2016*). These regions, however, were not observed for *MPs*. Finally, when testing the differential contrast *Target >MPs*, we confirmed an effect of the VS/Putamen in the direction of a stronger agreement-related activity for the *Target* (*Figure 5C*).

Finally, we tested whether discrepancy and agreement effects differed significantly across group (Controls vs. EMS). We did not found modulations at the level of neither dpMPFC/AI (for discrepancy) nor VS (for agreement). Unexpectedly, Controls showed different effects than EMS in the left Supramarginal Gyrus, for the *Target* feedback, and in MCC, for the *MPs* (see *Appendix 1—figure 2*, for more details).



**Figure 4.** Behavioral Results of Reapprisal and Distrust. (**A**) Boxplots and individual data describing the mean. Reappraisal index, resulted from the differential pain rating from two separate sessions (after vs. before presentation of feedbacks). Data are presented separately for Target (yellow dots)

*Figure 4 continued on next page*

*Figure 4 continued*

and MPs (violet dots) feedbacks, and for their relative position (Lower, Equal, Higher) with respect to participants initial ratings. The top subplot describes the data from Experiment 1, whereas the bottom subplot describes the data from Experiment 2. (B) For Experiment 1, we display also a scatter plot and confidence intervals area describing individual effects of MPs feedbacks' on Reappraisal plotted against Distrust (top row). Parameters (b) were obtained by a linear regression, similar to that used in the main analysis, but run separately for each individual subject. The higher b, the more the reappraisal is explainable according to the position of a specific feedback. The linear relation is further explored through boxplots displaying the Reappraisal for MPs' feedbacks, separately for individuals with high/low levels of distrust (bottom row). Note that in all subplots the feedbacks' position is displayed across three discrete categories to improve readability, although in the experiment it changed across a continuum. '***', '**', '*' refer to significance associated with paired t-tests or Spearman's ρ rank-correlation coefficient at p<0.001, p<0.01, and p<0.05 respectively.

## Effect of distrust

Consistently with what done in the analysis of pain expression ratings, we tested whether individual differences associated with the processing of feedbacks could be explained in terms of positive dispositions toward MPs (in Experiment 2). This analysis revealed no effects in behavioral ($F < 1.73$, $p>0.191$) or neural responses. We then tested for an effect associated with *Distrust* toward the facial expression's authenticity and found, in Experiment 1, a significant *MPs\*Distrust* interaction (Exp 1: $b = 0.04$, $F_{(1,3628.3)} = 6.58$, $p = 0.010$; Exp 2: $b = -0.02$, $F_{(1,74.88)} = 0.47$, $p = 0.496$). *Figure 3B* displays individual effects of *MPs* feedback on Reappraisal (obtained from a linear model fitted on each subject individually) increasing linearly with *Distrust*, thus suggesting that participants adjusted more their evaluation toward the physicians' feedback, the more they were suspicious on the expression's authenticity. Instead, Distrust played no role in the adjustment to the *Target* feedback or as a main effect ($F < 1.01$, $p > 0.317$). Unfortunately, we did not replicate the behavioral effects of distrust in Experiment 2. Interestingly, however, the analysis of the neural activity in Experiment two revealed that *Distrust* influenced how the Ventral Striatum (VS) responded to feedback agreement. This was observed when modeling *Distrust* against the differential contrast *Target >MPs*, which revealed that the effect observed in *Figure 5C* decreased linearly the more participants questioned the expression's authenticity. More specifically, whereas individuals with low distrust show strong agreement effect in this region, with enhanced activity whenever the *Target*'s feedback coincides with their own judgment, individuals highly doubtful about the expression's authenticity show comparable response for all *Target* feedback positions (*Figure 6B–C*, yellow dots). Furthermore, this effect did not generalize to the MPs' feedback, which exhibited the opposite trend (violet dots).

Finally, we carried out control analyses to assess whether the effects of *Distrust* were confounded by an overall lack of reliance toward the experimental paradigm (rather than the facial expression specifically). For this purpose, we reanalyzed *Distrust* in combination of another item form the post-experimental debrief assessing individual thoughts about another part of the task (the feedbacks). Full details are provided in Appendix 1, and confirm that participants' responses in our task were influenced selectively by considerations toward the authenticity of the facial expressions. Overall, our data show that *Distrust* influences the processing of the feedbacks about one's pain assessment, by inhibiting the neural processing implicated in treating information from the person in pain (Experiment 2), and consequently promoting adjustment toward the opinion of physicians (Experiment 1).

## Discussion

In the present study, we tested whether individuals on the verge of becoming physicians (experienced medical students; EMS) differed from younger colleagues and lay controls in the way in which they appraise people's pain, and take into account feedbacks from both the suffering person (*Target*) and other medical practitioners (*MPs*). For this purpose, we run two experiments (plus preliminary pilots): the first testing behavioral effects in students at different years of enrolment in medical faculty (plus lay controls), and the second recording neural activity (with fMRI) specifically on EMS and Controls. We found that EMS show decreased sensitivity toward painful facial expressions, in terms of both behavioral ratings and neural response in a predefined brain network (centered on AI and dACC) implicated in the processing people's sufferance (*Ding et al., 2020*; *Fan et al., 2011*; *Jauniaux et al., 2019*; *Kogler et al., 2020*; *Lamm et al., 2011*; *Timmers et al., 2018*). Instead, we found that the sensitivity to feedbacks was influenced by the degree of distrust. Indeed, individuals who were doubtful about the expressions' authenticity revealed decreased neural response in the



**Figure 5.** Surface rendering showing significant increase of neural activity associated with the (**A**) deviation or (**B**) agreement with the social feedbacks. Activation associated with the Target are displayed in yellow, whereas those associated with the MPs are displayed in violet. Common responses across the two kinds of feedbacks are displayed in light white. (**C**) Regions showing differential agreement effects with the Target and MPs feedbacks. All effects are displayed under a height threshold corresponding to p<0.001, with each region surviving cluster-correction for multiple comparisons for the

*Figure 5 continued on next page*

*Figure 5 continued*

whole brain. dpMPFC: dorsal-posterior Medial Prefrontal Cortex; vMPFC: ventral Medial prefrontal Cortex; TP: Temporal Pole; MFC: Middle Frontal Gyrus; IFG: Inferior Frontal Gyrus; AI: Anterior Insula; SPC: Superior Parietal Cortex; IOG: Inferior Occipital Gyrus; VS: Ventral Striatum. (D) Parameter estimates extracted from two regions of interest and displayed separately for Target (yellow dots) and MPs (violet dots) feedbacks, and for their relative position (Lower, Equal, Higher) with respect to participants ratings. Note that feedbacks' position is displayed across three discrete categories to improve readability, although in the experiment it changed across a continuum (see Materials and methods). '***', '**', and '*' refer to t-tests associated with $p<0.001$, $p<0.01$, and $p<0.05$, respectively.

reward system (Ventral Striatum [VS] and Putamen) when their judgment coincided with the self-report of the person in pain and were more incline to adjust their evaluation toward the opinion of other *MPs*. Overall, our study underscores the multiple processes influencing the evaluation of pain in medical practitioners, as early as in university years: whereas the students' seniority influenced the

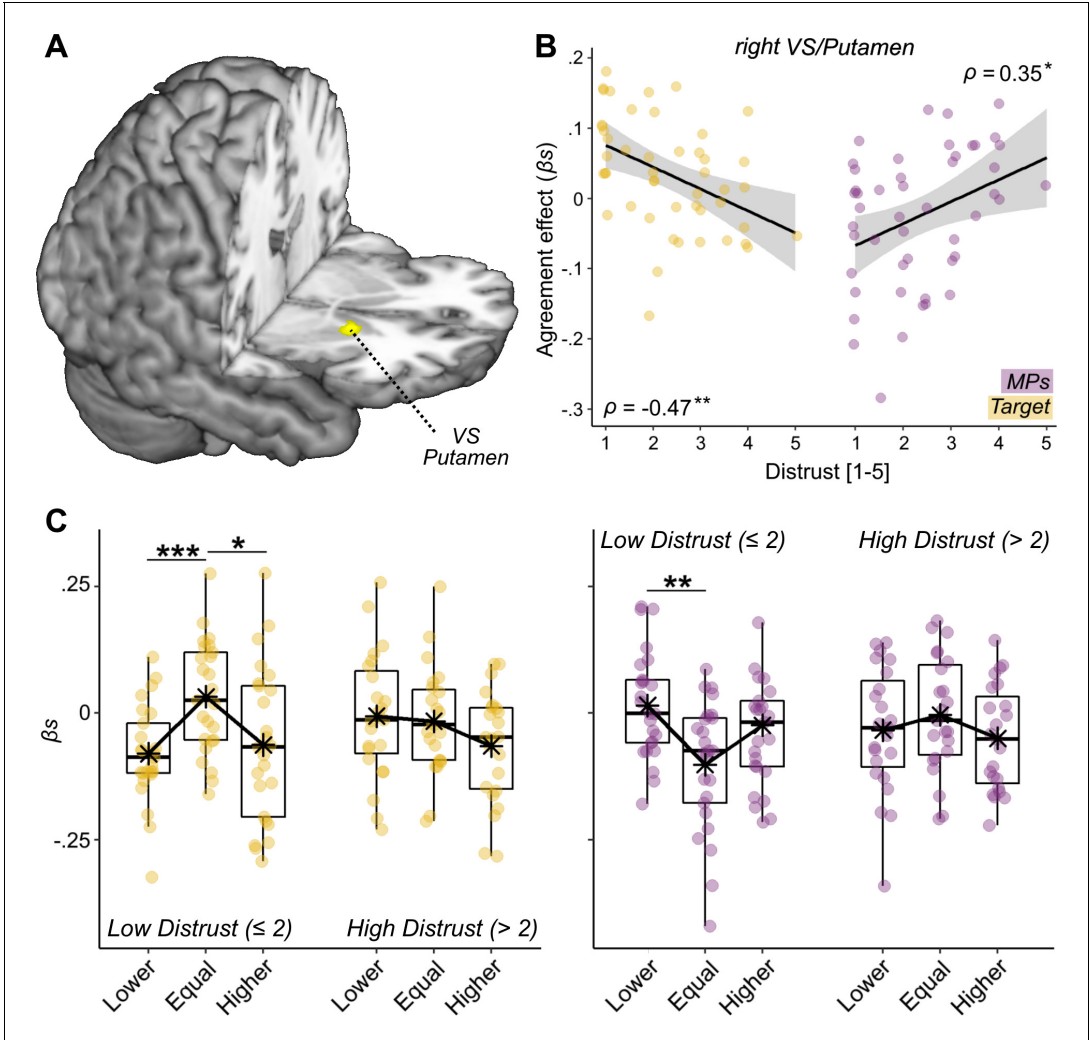

**Figure 6.** Distrust effect. (A) Surface rendering displaying the effects of agreement toward the Target feedback interacting with Distrust. The effect is displayed under a height threshold corresponding to $p<0.001$, and survivse cluster-correction for multiple comparisons for the whole brain. (B) The parameters extracted by the highlighted Ventral Striatum (VS) plotted against Distrust self-reports in a linear regression with confidence intervals area (higher parameters reflect stronger agreement effects). (C) The linear regression is further explored through boxplots displaying VS neural activity separately for Target (yellow dots) and MPs (violet dots) feedbacks, for their relative position (Lower, Equal, Higher) with respect to participants ratings, and separately for individuals with high/low levels of distrust. Parameter plots are associated with Spearman's ρ rank-correlation coefficient or paired-sample t-test, with '***', '**', and '*' referring to significance at $p<0.001$, $p<0.01$, and $p<0.05$, respectively.

assessment of the patient's pain, it is distrust that affected the way in which feedbacks from other people (including the patient) are taken into account.

## Experience in medical university influences pain processing

Consistently with previous studies showing that medical students (*Xie et al., 2018*) and practitioners (*Davoudi et al., 2008*; *Duignan and Dunn, 2008*; *Kappesser et al., 2006*; *Puntillo et al., 2003*; *Teske et al., 1983*) underestimated others' pain proportionally to their experience (*Choinière et al., 1990*; *Davoudi et al., 2008*), in Experiment 1 we found that students' seniority influenced negatively the rating of others' pain expressions (*Figure 2B*). Such effect appeared driven in large extent by young students who provide higher ratings than the other groups, whereas no difference was observed between Controls with no medical education and EMS (*Figure 2B–D*). This suggests that, at the beginning of their medical tenure, individuals might be highly vigilant to signs about patients' conditions (including pain), and that this effect decreased linearly in subsequent years.

Interestingly, Experiment two showed discrepancies between Controls and EMS in the neural response evoked from painful facial expressions. In particular, a network comprehending the anterior insula (AI) and dorsal anterior cingulate cortex (dACC) showed stronger linear effects of pain ratings in controls than in EMS (*Figure 3A*). This network has been repeatedly implicated in previous experiments involving the processing and empathizing with others' pain (*Ding et al., 2020*; *Fan et al., 2011*; *Jauniaux et al., 2019*; *Kogler et al., 2020*; *Lamm et al., 2011*; *Timmers et al., 2018*), and has been often considered the neural substrate for empathy (*Bernhardt and Singer, 2012*; *Lamm et al., 2019*; *Stietz et al., 2019*). Critically, previous studies have already shown how professional healthcare providers display decreased activity in these regions to the sight of others' injuries (*Cheng et al., 2017*; *Cheng et al., 2007*), an effect that is more pronounced in individuals with more experience in the field (*Cheng et al., 2017*). Our data extend these neural findings to a younger population at the end of their university program in medicine, thus suggesting a major role played by medical education in the responsiveness of this network. Furthermore, the analysis of brain activity of Experiment two allows to detect group differences (Controls vs. EMS) which were not observed from the behavioral measures from both experiments. This underlies the ability of neuroimaging techniques to capture also those neural processes which are too subtle to impact overt behavior.

The negative effect played by medical scholarly and professional experience in the neural response to others' pain has been often interpreted in terms of decreased empathic response, possibly promoted by enhanced regulatory abilities (*Cheng et al., 2007*; *Decety et al., 2010*), and allowing healthcare providers to interact with patients without any contagion from their sufferance (*Gleichgerrcht and Decety, 2014*; *Vaes and Muratore, 2013*; *Weilenmann et al., 2018*). Our data are not against this interpretation, although we advise caution in interpreting modulations in AI/ dACC to others' pain as reflective of overall empathy changes. Indeed, our groups did not differ in terms of empathic traits from dedicated questionnaires (see *Appendix 1—table 2*), and replacing the grouping variable with these scores in the neuroimaging analysis did not led to any result (scores of perspective taking influenced behavioral ratings, but only in Experiment 2). Also in the literature, positive correlations between AI/dACC response to others' pain and empathy scores were found in only ~30% of neuroimaging studies who tested this relation (see *Lamm et al., 2011*, for meta-analytic evidence). Furthermore, in contrast to neuroimaging findings, studies testing the role of medical experience on empathy scores have mixed results, with some suggesting a negative modulation (*Bellini and Shea, 2005*; *Hojat et al., 2009*; *Neumann et al., 2011*; *Smith et al., 2017*), and others reporting no change (*Cameron and Inzlicht, 2020*; *Xie et al., 2018*) or even a positive effect (*Handford et al., 2013*; *Kataoka et al., 2009*; *Smith et al., 2017*), with the direction of the effect changing as function of the score employed (*Smith et al., 2017*).

The mixed evidence from the literature could reflect the heterogeneity of the notion of empathy, which varies extensively across domains and scholars in terms of definitions and measures, with some tailored at measuring individuals' ability to share and resonate with pain specifically, whereas others tapping a more broad sensitivity to different emotional states (disgust, anger, joy) which are less relevant in healthcare settings. For these reasons, we prefer a more parsimonious interpretation, whereby medical experience impacts individual sensitivity to others' pain, rather than individuals overall empathetic abilities. This process has the beneficial effect of shielding physicians from the emotional weight sharing others' suffering (*Gleichgerrcht and Decety, 2014*; *Vaes and Muratore,*

2013; *Weilenmann et al., 2018*), but at the same time exposes patients (especially women and individuals from ethnical minorities, *Ghoshal et al., 2020*; *Greenwood et al., 2018*; *Kaseweter et al., 2012*; *Todd et al., 2000*) to the risk of being unrelieved from their condition.

## General and specific effects in feedback processing

When analyzing the degree to which individuals react to feedbacks, we found evidence of both general effects (common to *Target* and *MPs*), but also modulations specific for one source of information. At the behavioral level, we found that participants adjusted their pain ratings as function of any kind of feedback (*Figure 4*). This was mirrored by the analysis of brain activity, which revealed that the signal in a widespread network centered in dpMPFC increased linearly with the absolute discrepancy between participants' initial ratings and the presentation of any kind of feedback (*Figure 5A*). These converging effects of discrepancy for *Target*/*MPs* were highly reminiscent to those observed in previous studies testing social discrepancy in other forms of perceptual judgment (*Wu et al., 2016*; *Appendix 1—figure 1*).

To the best of our knowledge, there are three alternative ways to interpret these general effects. First, part of participants' monitoring system treats information from the physicians as relevant as that from the person experiencing pain directly. Hence, although the feedbacks are interpreted in terms of their social meaning, none of them are taken into account in a privileged fashion, at least in the context of discrepancies. Second, it is possible that participants were captured by the presence of any visual stimulus far from their current attentional focus (the initial rating). This conjecture draws from a recent study employing a similar paradigm, in which participants' reappraisal was also influenced by feedbacks with no social meaning (*Kim and Hommel, 2015*, but see *Klucharev et al., 2009*). Third, scholars suggested a potential confound of the initial rating, which depends on the position of the feedback (*Ihmels and Ache, 2018*; *Kim and Hommel, 2018*). Indeed, trials with large deviance in one direction are by definition those in which the initial rating is sufficiently distant from the physical boundary of the scale to allow the presentation of the feedback (see methods). Hence, discrepancy effects could (at least in principle) be a delayed after-effects of the previous rated pain. Currently, our results do not allow to dissociate between these three alternative interpretations. Future research, involving the implementation of additional control conditions, will ascertain whether the divergence effects observed here underlie a true processing of the social information conveyed by the feedbacks.

In sharp contrast from discrepancy effects, the neural modulations evoked by agreement appear to be selective for one source of information. More specifically, in line with previous studies investigating effects of social influence (*Klucharev et al., 2009*; *Wu et al., 2016*), we found that the agreement with the *Target* feedback enhanced activity in the VS and Putamen (*Figure 5B*). Critically, such effect was not observed for *MPs* feedback, a pattern confirmed also when testing the direct contrast *Target* >*MPs* (*Figure 5C*). Hence, the analysis of convergences provides clear evidence that participants do take into account the social meaning of the feedbacks (over and above other potential non-social confounds), with VS/Putamen privileging information arising from the person in pain as opposed to that coming from physicians. Furthermore, to the best of our knowledge, this is the first study documenting a dissociation of different networks subtending divergence *vs.* agreement effects in social influence paradigms. Indeed, *Klucharev et al., 2009* interpreted the neural responses in social influence paradigms as part of a unique mechanism for reinforcement learning, according to which midbrain dopaminergic signals monitor whether an outcome is perceived as negative or positive by projecting to neural structures in dpMPFC and VS, respectively (*Klucharev et al., 2009*). Within this framework, divergence/convergence effects are essentially two sides of the same coin. However, this view is not supported by our data showing a further functional differentiation among the two sets of regions. Whereas discrepancy effects appear to work on a more general level (without distinguishing between different sources of information), agreement effects in VS/Putamen were specific for the feedback from the person in pain, suggesting a selective monitoring of the most relevant source of information for the decision. Hence, VS is not merely the opposite of dpMPFC, but responds according to different principles/priorities.

## Distrust influences specific feedback processing

Across both experiments, we found that *Distrust* was the key factor explaining whether participants valued in greater extent the information arising from *Target* or *MPs*. In Experiment 1, participants most doubtful about the pain's authenticity adjusted their response more strongly toward *MPs*. This effect was not replicated in Experiment 2 (in which we recruited a smaller sample size, and did not include young and intermediate medical students). However, in Experiment 2, *Distrust* influenced the neural responses at the level of VS, with agreement effects for the *Target* decreasing progressively the more participants doubted the faces in the videos, and agreement effects for *MPs* showing the opposite trend. Despite their differences, the two studies converge in pointing a major role of *Distrust* in the way in which individuals process feedbacks about their pain evaluations. Importantly, such effect is not confounded with participants' overall distrust toward the experimental set-up, but reflect specific considerations of the facial expressions implemented (see Appendix 1). Furthermore, *Distrust* operates independently from the grouping factor and from the personal dispositions toward the category of doctors (measured through the IAT). Hence, our effects might not reflect experience, belongingness or personal positive dispositions toward *MPs*, but possibly the fact that doctors are figures of authority which become much more salient when the facial expression cannot be relied upon.

Previous studies already revealed that healthcare providers underestimated patients' pain in greater extent when they were given hints about potential deceptive attempts to get attention or non-necessary medication (*Kappesser et al., 2006*). Furthermore, qualitative investigations suggested that emergency nurses sometimes have difficulty suspending their point of view when confronted with patients whose pain-reports conflict strongly with their initial judgment (*Johannessen, 2019*; *Vuille et al., 2018*). Our study provides a mechanistic explanation of these observations, by revealing that distrust operates on part of the network for feedback-based learning, and specifically by altering the sensitivity of the reward system to information from the person in pain. Plainly put, if the *Target* is not to be trusted, his/her self-reports become less relevant, and VS might react less strongly to any agreement with one's rating.

Our findings are particularly relevant in clinical settings, where pain appraisal is the basis for the selection of subsequent therapeutic procedures, including the prescription of strong (and potentially dangerous) painkillers. In this context, the deontological need to relieve patient's pain is often counterweighted by the equally relevant need to prevent future side-effects and complications, a conflict that is often resolved by each individual based on personal ability to cope with uncertainty and sensitivity to errors (*Corradi-Dell'Acqua et al., 2019*). Obviously, a key source of information for these decisions is the reliability of available pain cues (e.g. *is the facial expression genuine?*) which, if estimated below a given threshold, can relieve caregivers from any struggle by prioritizing other clinical considerations over the management of patients' pain. As unfortunate drawback, estimated reliability might be vulnerable to biases, including those related to the social and ethical condition of the patient. Previous studies showed that confronting individuals from different social and ethnic group can impact profoundly representation of trust and associated activity in VS (*Hughes et al., 2017*; *Stanley et al., 2012*; *Stanley et al., 2011*), but also sensitivity to their pain (*Avenanti et al., 2010*; *Mende-Siedlecki et al., 2019*) including the neural responses in AI and dACC (*Cao et al., 2015*; *Hein et al., 2010*; *Xu et al., 2009*). As such, the interplay between the reward system and AI-dACC network might offer a plausible model to explain why specific social/ethnic categories are less likely to be acknowledged for their pain and receive adequate treatment. Future studies will need to explore this.

## Limitations of the study and concluding remarks

The present study investigates the role of medical experience by recruiting independent groups at different years of university. As such, our study shares the weaknesses of cross-sectional investigations (*Wang and Cheng, 2020*), as the role of experience was not tested longitudinally in the same population. In particular, some of our effects might be influenced by individual traits and features which are more frequently observed in medical students (especially those who reach the end of their tenure) as opposed to lay controls. In our analysis, we took care to account for several potential confounding variables, such as gender, age, or empathic traits. Other measures (e.g. personality traits) appeared fairly stable across the different groups (with only few differences, not systematic across

experiments; see *Appendix 1—table 2*), and therefore are unlikely to have confounded our results. Yet, it is still possible that our results could have been influenced also by other factors untested in the present study. For instance, relevant missing information are the detailed educational level of Controls (including their year of enrollment at university) and participants' ethnic group. Finally, in designing our study we followed previous paradigms in which social feedbacks (in our case *MPs*) were described as the aggregate opinion of many individuals from a community (*Klucharev et al., 2009*; *Koban and Wager, 2016*). This however opens the question as to whether *Distrust* promotes reliance toward *MPs* feedback due to their professional background, their size (an opinion of 20 is more reliable than that of one) or a combination of both factors. Future studies will need to duly account for these limitations.

Notwithstanding these considerations, our study provides compelling evidence of an independent contribution of experience and *Distrust* in different aspects of pain management. Experience in medical faculty influenced individuals' assessment of pain, both at the level of explicit ratings, and on a network comprehending insular and cingulate cortex knowingly sensitive to cues about people's suffering (*Ding et al., 2020*; *Fan et al., 2011*; *Jauniaux et al., 2019*; *Kogler et al., 2020*; *Lamm et al., 2011*; *Timmers et al., 2018*). Additionally, *Distrust* affected the processing of feedbacks about one's assessments, by decreasing striatal activity whenever the *Target*'s reports match participants' initial ratings, and by leading participants to rely more on the opinion of other doctors. Overall, our study documents relevant changes in the neural circuity underling pain assessment and feedback processing that occur already at the very early stages of medical career.

## Materials and methods

### Experiment 1

#### Participants

Because this was an exploratory experiment, we recruited an overall of 120 participants, organized in four groups of 30 subjects. The first group comprehended lay individuals (12 males out of 30, mean age = 25.03 years±4.75 Standard Deviation [SD]) who were recruited among different faculties and professions, except those related to medicine, infirmary, dentistry, and physiotherapy. The remaining 90 participants (26 males, mean age = 22.56 ± 2.86) were recruited among students enrolled from the 1 st to the 6th year of medical faculty at the University of Geneva and Lausanne (1 st year = 14, 2nd year = 16, 3rd year, N = 15; 4th year, N = 15; 5th year, N = 17; 6th year, N = 13). We divided the total sample into additional groups of 30 participants each: controls, Young Medical Students (YMS) from the 1st and 2nd year of faculty, Intermediate Medical Students (IMS) from the 3rd and 4th year and Experienced Medical Students (EMS) from the 5th and 6th year. The participants did not report any history of neurological nor psychiatric illness and were naïve to the purpose of the study. Furthermore, they signed an informed consent prior to the experiment. This research (both experiments) was conducted in accordance with the Declaration of Helsinki and was approved by the local ethical committee (Commission Cantonale d'Éthique e de la Recerche [CCER] of Geneva, protocol code: CCER N. 2016–01862).

#### Stimuli

We used a database of 44 videos of painful facial expressions from *Lamm et al., 2007* depicting Caucasian actors wearing headphones and simulating painful reactions to a high pitch noise (see *Figure 1*). A validation pilot experiment (Pilot 2) on 24 independent lay participants (seven males, age 23.54 ± 4.12) revealed that the pain of these expressions was rated on average 7.42 ± 1.26 on a scale ranging from 1 (no pain at all) to 10 (worst pain imaginable). Ratings for each isolated video were highly similar with one another, with the least painful clip evaluated on average 7.01 ± 1.55 and the most painful clip 7.88 ± 1.41.

#### Experimental set-up

The main experimental session was organized in three blocks. In the first block, participants saw a fixation cross (1 s) followed by one video (3 s) and were asked to estimate the amount of pain experienced by the person in the video, by moving a randomly-presented rectangular cursor on a Visual Analogue Scale (VAS) ranging from 'no pain at all' (corresponding to value 1) to 'the worst pain

imaginable' (corresponding to value 10). Following the rating (7 s), participants were exposed to two simultaneous feedbacks (3 s) that represented the judgement of the *Target* of the video and the average judgement of 20 emergency doctors that previously rated the same video-clips. These two feedbacks were displayed by cursors similar to that of the participant but of different colors (color codes were randomized across participants). Each kind of feedback was independent and orthogonal as respect to the other and described pain levels, which could be higher, equal or lower, compared to the participant. This led to a 3 × 3 design, with nine different conditions. Feedbacks were located on the scale according to a normal distribution with mean +2.2, 0 and −2.2 scale points for values higher, equal and lower ratings respectively, and ±0.5 as standard deviation that helped in providing an effect of variability consistent with that of a human judgement. In case the feedbacks had to appear out of the physical limit of the scale, they were instead displayed at the further extremity within the scale. Each video appeared just once (total number = 44) and was randomly assigned to one of the nine conditions. This led to eight conditions associated with five videos, and one condition associated with four videos (5*8 + 4 = 44). The association between conditions and videos was different for each subject, thus preventing the presence of one condition systematically associated with less videos.

In a second block, participants completed a battery of questionnaires in a time-window of 30 min (see below). This block served also to the purpose of allowing some time to pass between the first and the third block. If participants completed the questionnaires before the end of the 30-min session, they were asked to wait for the remaining time. If they were still filling the questionnaires at the end of the 30th minute, they were asked to discontinue and finish after the third block. This insured that for all participants the same delay was implemented between the first and third block. The questionnaires comprehended the French versions of the Interpersonal Reactivity Index (IRI) (*Davis, 1980*); the Situation Pain Questionnaire (SPQ) (*Clark and Yang, 1983*); the Pain Catastrophizing Scale (PCS) (*Sullivan et al., 1995*) and the Big Five Inventory (BFI) (*Plaisant et al., 2010*). See *Appendix 1—table 2* for more details.

Finally, in the third block participants were unexpectedly asked to rate again all 44 videoclips, without however being exposed to any feedback. The experiment was programmed using Cogent (Wellcome Dept., London, UK) toolbox as implemented in Matlab R2012a (Mathworks, Natick, MA).

## Procedure

After signing the informed consent, participants could read the instructions and familiarize with the association between the cursors' colors and their meaning. During the rating session, responses were collected by pressing highlighted keys on the computer keyboard. The overall experiment lasted approximately 75 min (block 1:~15; block 2: 30 min and block 3:~15), and was followed by a short debriefing session in which participants replied to ad hoc questions testing several aspects of the experimental manipulation (see Appendix 1 for further details). Finally, participants were asked to complete a set of questionnaires that they started during block two in case they could not finish before 30 min. The experiment took place at the Brain Behavioral Laboratory (BBL) in the Centre Médical Universitaire (CMU) of the University of Geneva, Switzerland.

## Data analysis

As a first step, we analyzed if the four groups of interest changed as function of their rating from the first session. This was achieved by feeding the median pain ratings in an Analysis of Variance with *Group* as between-subject factor. To account for potential age differences between the group, the ANOVA was also repeated by using age as covariate.

Subsequently, we assessed whether participants changed their initial ratings as function of the social feedback. Following previous studies in the field (*Klucharev et al., 2009*; *Koban and Wager, 2016*), a measure of Reappraisal was considered, defined as the differential rating between the third and first block for each video. Single trial Reappraisal values were fed to a Linear Mixed Model (LMM) with the relative position of the *Target* and *MPs* feedbacks as continuous predictors. Furthermore, we also included *Group*, *Gender*, and *Age* as between-subject predictors, each modeled as both main effect and in interaction with *Target* and *MPs*. Finally, the identity of the subjects and of the individuals portrayed in the videos was modeled as random factor (with random intercept, and slope for *Target* and *MPs*). Additionally, we assessed whether individual differences in the task could

be well explained in terms of beliefs about the authenticity of the pain observed. To assess this, the same LMM described above was repeated by replacing the *Group* factor with *Distrust* scores as continuous between-subject predictor. This score was obtained by the combination of two post-experimental debrief items (see Appendix 1 for more details). *Item 1*: 'I had the impression that the people in the video were simulating unpleasantness' (in French: 'J'ai eu l'impression que les gens de la vidéo ont simulé le désagrément'); *Item 2*: 'I had the impression that the pain observed was real' ('J'ai eu l'impression que la douleur observée était réelle'). For both items, participants could give an answer in a five points Likert scale ranging from 1 [not at all] to 5 [absolutely]. As the reports of these two debrief items are complementary, a combined measure was calculated as follows: $(Item1 + (6 - Item2))/2$, which describes a *Distrust* scalar ranging from 1 (not at all) to 5 (extremely). The significance of the relevant effects in the LMM was calculated using the Satterthwaite approximation of the degrees of freedom, as implemented in the lmerTest package (*Kuznetsova et al., 2015*) from R.3.4.4 software (https://cran.r-project.org/).

## Experiment 2

### Participants

For this experiment, 28 experienced medical students (EMS) enrolled to the 5th and 6th year of medicine at the University of Geneva, responded to our recruitment call. Two of them were excluded due to non-agreement with the task. Therefore, the final sample of EMS was composed by 26 participants (nine males, age = 24.15 ± 1.40). EMS participants were associated with a matched control group of 26 students not enrolled in medical-related faculties (13 males, age = 23.73 ± 4.11). The recruitment was carried out as for Experiment 1.

### Experimental set-up and procedure

After having read and signed the consent form and MRI security check-list, participants lay supine on the scanner with their head fixated by firm foam pads. They underwent a unique scanning session of about 20 min, where they carried out the same task as in Experiment 1-block 1, with few changes. Firstly, due to exigencies of the functional data acquisition in the MRI, the inter-stimulus interval was jittered and ranged from 2 to 5 s. Furthermore, the two social feedbacks were not rectangular shapes, like in the case of the previous experiment, but two little icons representing schematically a physician (a shape of a person with a stethoscope) and the person depicted in the video (a shape of a person with headphones; consistently with the video database implemented, *Figure 1*). This change was motivated by the fact that the rectangular feedbacks used in Experiment 1 (*Figure 1*) could have been easily confused. Although all participants from Experiment one recalled the association between the rectangle color and its meaning in the debrief, it is still possible that, in several instances during the task, the different meaning of the two feedbacks might have slipped participants' mind. Hence, replacing the rectangular shapes with icons transparently interpretable should have minimized any risk of confusion, and provide a more sensitive experimental setting. A behavioral pilot study (Pilot 3) on 23 Control students (11 males, 22.26 ± 3.86 years old), confirmed that this modified set-up led to the similar effects of Reappraisal than Experiment 1 (main effect of *Target*: b = 0.07, $F_{(1,26.60)}$ = 3.09, p=0.081; *MPs*: b = 0.15, $F_{(1,20.17)}$ = 7.08, p=0.015). Furthermore, when comparing the data from this pilot with those from the Controls from Experiment 1, no effect of Experiment was found, neither as main effect, nor as interaction with *Target/MPs* ($F < 1.22$, p>0.270). Hence, the modified set-up had no impact in Controls' behavioral responses.

Following the main task, participants exited the scanner, and filled demographic for about 30 min. Subsequently, as for Experiment 1, they were unexpectedly asked to rate again all the video-clips again, without being exposed to any feedback. Subjects carried out an Implicit Association Task identical to that of Pilot 1, and finally were debriefed. The whole experiment took place at the Human Neuroscience Platform of the Campus Biotech in Geneva.

The visual stimuli were presented on a 23' MRI compatible LCD screen (BOLDScreen23; Cambridge Research Systems, UK), at a resolution of 1024 × 768 (refresh rate 60 Hz) subtending a visual angle of approximately 11.8° (vertical) x 15.6°. Key-presses were recorded on an MRI-compatible bimanual response button box (HH-2 × 4 C; Current Designs Inc, Philadelphia, PA).

## Data analysis

Behavioral data from Experiment two were organized and analyzed in the same way like that of Experiment 1.

## Brain activity

As for the neuroimaging data, these were acquired through a Siemens Magnetom Prisma 3T scanner with a 64-channel head-and-neck coil. We used a multiband sequence with TR = 1100 ms, (TE) = 32 ms, flip angle = 50˚, 66 interleaved slices, 112 × 112 in-plane resolution, 2 × 2×2 mm voxel size, no inter-slice gap, multiband acceleration factor 6. Field map was estimated through the acquisition of 2 functional images with a different echo time (short TE = 4.92 ms; long TE = 7.38 ms). Structural images were acquired with a T1-weighted MPRAGE sequence (192 slices, TR = 2300 ms, TE = 2.32 ms, flip angle = 8˚, slice thickness of 0.9 mm, in-plane resolution = 256 × 256, 0.9 × 0.9 × 0.9 mm voxel size).

Functional images were fed to standard preprocessing pipeline as implemented in SPM12 software (Wellcome Department of Cognitive Neurology, London, United Kingdom). This pipeline did realignment, unwrapping (using a field map image to account for geometric distortions because of magnetic field inhomogeneity), and normalization to the 152 Montreal Neurological Institute (MNI) with voxel size resolution of 2 × 2×2 mm. Finally, the volumes were smoothed by convolution with an 8 mm full width at half maximum Gaussian kernel.

The preprocessed images of each individual were then fed to a first level analysis using the general linear model framework implemented in SPM. More specifically, we modelled the onset of each pain expression as events of 3 s (corresponding to the duration of the video-clip). We also assessed whether face-evoked activity was explained by the pain ratings that participants were about to make through a dedicated parametrical modulator. Subsequently, we modelled the occurrence of the VAS as events of 7 s (corresponding to the delay between the presentation of the scale and the subsequent feedbacks). For this condition, we also modeled the actual displacement carried out by participants (the absolute difference between the randomly presented initial position of the cursor and participants' final rating) as a parametric modulator. Finally, and more critically, we modeled the occurrence of the feedbacks as events of 3 s. Feedback-related activity was associated with two parametric modulators: first the absolute deviation between participants' rating and the position of the *Target*'s feedback, and the absolute deviation between participants' rating and the position of *MPs* feedback. Please note that, within this model, positive effects of each of these parametric modulators should be interpreted as brain activity increasing linearly with feedbacks' divergence from participants' initial rating. Instead, negative effects should capture effects related with feedbacks' convergence with the participants' rating. For each of the three main events vectors (facial expressions, VAS, and feedbacks), we accounted for habituation effects in neural responses by using the time-modulation option implemented in SPM, which creates additional regressors in which the trial order is modulated parametrically. This led to an overall of 10 regressors (three conditions + 7 parametric modulators), which were convolved with a canonical hemodynamic response function and associated with regressors describing their first-order time derivative. To account for movement-related variance, physiological-related artifacts, and other sources of noise, we also included the six realignment parameters, an estimate of cardiac- and inspiration-induced changes in the BOLD signal based on PhysIO toolbox (*Kasper et al., 2017*), and the average non-gray matter signal, defined as the coordinates with grey matter tissue probability <0.02. This is the largest non-grey matter mask whose average signal is not confounded by the task manipulation (see *Appendix 1—table 7*), and as such represents an estimate of for global sources of noise that might can be accounted for without biasing the results (*Aguirre et al., 1998*; *Desjardins et al., 2001*; *Junghöfer et al., 2005*). Low-frequency signal drifts were filtered using a cutoff period of 128 s. Serial correlations in the neural signal were accounted through exponential covariance structures, as implemented in the 'FAST' option of SPM.

Parameters associated with conditions of interest (e.g. parametric modulators of *Target*, *MPs* or contrast testing the differential effect between the two) were then fed in a second level, independent-samples t-test using random-effect analysis. Furthermore, inter-individual differences related to the IAT D-Score or to distrust estimates were modelled through linear regression. In all analyses, *Gender* and *Age* were always modeled as nuisance regressors. As first step, we considered only

those effects throughout the whole brain that exceeded p<0.05, family-wise correction for multiple comparisons at the cluster level (*Friston et al., 1994*), with an underlying height threshold of p<0.001, uncorrected. As second step, we also restricted our hypothesis based on regions previously implicated in the same paradigms. Specifically, for the processing of painful expressions, we took the most recent meta-analysis on pain empathy from *Kogler et al., 2020*, and selected the regions jointly implicated in processing the sight of others' suffering and one's own pain response (Figure 15 in *Kogler et al., 2020*, freely available at https://anima.fz-juelich.de/studies/Kogler_Empathy_2020). Instead for the processing of social feedback, we took the meta-analysis maps from *Wu et al., 2016* revealing differential effects for: (1) divergence from social feedbacks, implicating a network containing mPFC and right AI; (2) convergence with social feedbacks, implicating VS (see Table 3 in *Wu et al., 2016*; provided by the study's author). In both cases, the meta-analytic activation maps (thresholded to survive FWE correction for the whole brain) were binarized and used for small volume correction in our study. Within these areas, we considered significant those effects associated with p<0.05 FWE small volume correction at the voxel level.

## Acknowledgements

This research was financed by the Swiss National Science Foundation grants n PP00O1_157424 and PP00P1_183715 awarded to CCD. GD was also supported by the Ernst and Lucie Schmidheiny Foundation. We thank Asli Erdemli for the assistance with the acquisition of the last participants, and Chunliang Feng for sharing with us the activation maps from his meta-analysis (*Wu et al., 2016*). Finally, we thank Roberto Martuzzi and Loan Mattera for their assistance in all matters involving MRI scanning.

## Additional information

### Funding

| Funder | Grant reference number | Author |
| --- | --- | --- |
| SNSF | PP00P1_157424 | Corrado Corradi-Dell'Acqua |
| SNSF | PP00P1_183715 | Corrado Corradi-Dell'Acqua |
| Fondation Ernst et Lucie Schmidheiny | | Giada Dirupo |

The funders had no role in study design, data collection and interpretation, or the decision to submit the work for publication.

### Author contributions

Giada Dirupo, Conceptualization, Data curation, Formal analysis, Visualization, Methodology, Writing - original draft, Project administration, Writing - review and editing; Sabrina Totaro, Jeanne Richard, Project administration; Corrado Corradi-Dell'Acqua, Conceptualization, Supervision, Funding acquisition, Writing - original draft, Writing - review and editing

### Author ORCIDs

Giada Dirupo ⬤ https://orcid.org/0000-0003-4578-6943
Corrado Corradi-Dell'Acqua ⬤ http://orcid.org/0000-0002-7512-9023

### Ethics

Human subjects: all subjects read and signed an informed consent prior to taking part to the experiement, thus agreeing that their data could be published under anonimity. They had the time to read and ask for clarification/ information the the researcher conductiong the experiment in case they wanted to. This research was conducted in accordance with the Declaration of Helsinki and was approved by the local ethical committee (Commission Cantonale d'éthique e de la Recherce [CCER] of Geneva, protocol code: CCER N. 2016-01862).

Decision letter and Author response
Decision letter https://doi.org/10.7554/eLife.63272.sa1
Author response https://doi.org/10.7554/eLife.63272.sa2

## Additional files

### Supplementary files

• Transparent reporting form

### Data availability

The behavioral data and script are stored and available at the following link: https://osf.io/qnp6m/
The UPDATED (revision1) brain imaging data are stored and available at the following link: https://neurovault.org/collections/9006/.

The following datasets were generated:

| Author(s) | Year | Dataset title | Dataset URL | Database and Identifier |
|-----------|------|---------------|-------------|-------------------------|
| Dirupo G | 2021 | Medical education and distrust modulate the response of insular-cingulate network and ventral striatum in pain diagnosis. | https://osf.io/qnp6m/ | Open Science Framework, qnp6m |
| Dirupo G | 2021 | Brain networks for pain diagnosis. Differential contribution of medical education and distrust in the appraisal of others' pain | https://neurovault.org/collections/9006/ | NeuroVault, 9006 |

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

## Appendix 1

### Debrief questions

The main experiment involved three experimental sessions, which were followed by a short debriefing where participants replied to ten *ad hoc* questions testing several aspects of the experimental manipulation. These were

1. I had the impression that the people in the video were simulating unpleasantness (the original French formulation was: *J'ai eu l'impression que les gens de la vidéo ont simulé le désagrément*)
2. I had the impression that the pain observed was real (*J'ai eu l'impression que la douleur observée était réelle*)
3. I had the impression that the doctor often judged the stimulus as being less unpleasant than I did (*J'ai eu l'impression que le médecin souvent juge le stimulus comme moins désagréable en comparaison à moi*)
4. I had the impression that the protagonist of the video often judged the stimulus as being less unpleasant than I did (*J'ai eu l'impression que le protagoniste de la vidéo souvent juge le stimulus comme moins désagréable en comparaison à moi*)
5. I had the impression that the doctor often judged the stimulus as being more unpleasant than I did (*J'ai eu l'impression que le médecin souvent juge le stimulus comme plus désagreable en comparaison à moi*)
6. I had the impression that the protagonist of the video often judged the stimulus as being more unpleasant than I did (*J'ai eu l'impression que le protagoniste de la vidéo souvent juge le stimulus comme plus désagréable en comparaison à moi*)
7. I had the impression that the judgment from either protagonist or doctor judgments were not real (*J'ai eu l'impression que le jugement soit du protagoniste de la vidéo soit du médecin n'etait pas réel*)
8. I had the impression that this experiment was built to see if I changed judgment the second time (*J'ai eu l'impression que cette experience était faite pour comprendre si j'allais changer mon jugement dans la deuxième partie*)
9. I had the impression of having changed my judgment the second time (*J'ai eu l'impression d'avoir changé mon jugement dans la deuxième partie*)
10. more towards the protagonist or the doctor? (*plus vers le médecin ou le protagoniste de la vidéo?*).

Items 1–9 were answered by selecting a number on a Likert scale ranging from 1 (not at all) to 5 (absolutely). Instead item 10 was answered by selecting one of two options: the Protagonist of the video (i.e. *Target*) and *MPs*.

### Implicit Association Task

The study was carried out under the assumption that medical students present more pronounced positive dispositions towards physicians, the more they progressed across their medical tenure. This assumption was tested through an independent pilot online experiment (Pilot 1) carried out on 155 subjects (41 males, mean age = 21.85 ± 2.81), divided into 29 Controls, 56 YMS, 32 IMS and 38 EMS. These carried out an online Implicit Association Task (IAT) (*Greenwald et al., 1998*), aimed at detecting the mental association between positive/negative terms (hereafter categories) and pictures of MPs or Bankers (hereafter target pairs). For this task, positive and negative words were taken from the French version of *Project Implicit* of Harvard (https://implicit.harvard.edu/implicit/), whereas the target pairs were half-body photos of people wearing either white coats (MPs) or a suit (bankers), balanced for gender. The task was composed by seven blocks, each preceded by instructions. The experiment was designed and carried out using the *iatgen* tool (https://iatgen.wordpress.com/) on the online software *Qualtrics* (https://www.qualtrics.com/fr/), under default parameters. More specifically, the task was organized into 6 blocks of 20 trials each, where individuals were asked to classify either targets or categories by pressing one of two keyboard keys. In each trial, a fixation cross appeared for 250 ms in the middle of the screen, followed by a target/category which remained visible until a keypress was provided. In case a sorting errors, a red cross was superimposed to the stimulus for 300 ms. At that point participant could press the correct key.

In the two main conditions, participants classified a category-target association. In the 'positive' blocks they had to press one key for the occurrence of MPs or positive words, and the other key for bankers or negative words. In the 'negative' blocks, they had to press one key for the occurrence of MPs or negative words, and the other key for bankers or positive words. Individuals positively-biased towards MPs should be faster when positive words were mentally associated with MPs pictures, as opposed to Bankers. In this perspective, the average response times between the 'negative' blocks was subtracted with that of 'positive' blocks. The resulting differential value was normalized by the pooled standard deviation of the two blocks (see default settings from https://iatgen.wordpress.com/, *Greenwald et al., 1998*, for more details). The higher this D-score parameter, the stronger the association between positive valence and MPs.

An analysis of variance (ANOVA) with the D-score as dependent variable and Group, Gender and Age as independent variables revealed only a main effect of Group ($F_{(3,148)}$ = 3.19; p=0.025; all other effects: $F < 0.51$; p>0.475). T-test post-hoc analyses showed a difference between EMS and Controls ($t_{(65)}$ = 2.75; p = 0.008), and IMS and Controls ($t_{(59)}$ = 2.19; p=0.032, see *Figure 2A*). No significant difference was found when testing all the other possible combinations of groups ($|t| \leq 1.95$, p$\geq$0.054).

## Follow up analyses on Distrust

In the present study we found how participants' responses in the main task were influenced by their *Distrust* towards expressions' authenticity, as measured in an *ad hoc* debrief questionnaire. Here we run follow-up analyses to test whether these effects were confounded by an overall lack of reliance towards the experimental manipulation, rather than the facial expressions specifically. For this purpose, we analyzed *Distrust* in combination with another post-experimental debrief measure: 'I had the impression that the judgment from either protagonist or doctor judgments were not real' (see Appendix 1 for all debrief questions). Like for the case of *Distrust*, also the scores from this question reflect participants' doubts about some aspects of the paradigm. However, in this case participants did not focus on the video clips, but rather on the delivery of the feedback. Interestingly, scores from this item (hereafter *ExpDistrust*) correlated significantly/marginally with those of *Distrust* (Exp. 1: Spearman's $\rho$ = 0.15, p=0.097; Exp. 2: $\rho$ = 0.42, p=0.003) pointing to a common source of variance.

In the present study, Distrust was implicated in the behavioral responses of Experiment 1. More specifically, pain ratings were negatively influenced by *Distrust* ($b = -0.31$, $F_{(1,113)}$ = 5.50, p=0.021), whereas the analysis of Reappraisal was significantly associated with a *MPs*Distrust* interaction ($b = 0.04$, $F_{(1,3628.3)}$ = 6.58, p=0.010 – *Figure 4b*). We then repeated the same analyses by replacing the scores of *Distrust* with those from *ExpDistrust*, in order to assess whether this new predictor led to similar results. In none of the analyses we found any effect associated with *ExpDistrust*. More specifically, we found no negative modulation in the analysis of pain ratings ($b = -0.01$, $F_{(1,113)}$ = 0.005, p=0.941), and no *MPs*ExpDistrust* interaction in the analysis of Reappraisal ($b = 0.003$, $F_{(1,1329.3)}$ = 0.08, p=0.771). Subsequently, we repeated the analyses by modeling *Distrust* and *ExpDistrust* together, with the former used as a variable of interest, and the latter as a nuisance covariate. Within this setting, we confirmed all results associated with *Distrust*, both in the analysis of pain ratings ($b = -0.32$, $F_{(1,112)}$ = 5.46, p=0.021), and that of Reappraisal (*MPs*Distrust*: $b = 0.04$, $F_{(1,3513.8)}$ = 6.55, p=0.011).

*Distrust* was also implicated in the analysis of neural response in Experiment 2. More specifically, we found that the VS activity evoked by *Target* agreement decreased linearly with *Distrust* (local maxima: x = 0, y = 10, z = 4, $t_{(44)}$ = 3.55, *p* (uncorrected) <0.001, *p* (small volume correction) = 0.021 – see *Appendix 1—table 4*). The same region was implicated in the interaction between the source of the feedback (*Target vs. MPs*) and *Distrust* (local maxima: x = 14, y = 20, z = 6, $t_{(44)}$ = 4.90, *p* (uncorrected) <0.001, Cluster size = 277, *p* (whole-brain correction at the cluster level) = 0.003 – see *Figure 6* and *Appendix 1—table 6*). We then repeated the same analyses by replacing the scores of *Distrust* with those from *ExpDistrust*, and found no significant effects, neither under whole-brain correction, nor under small volume correction searching within a predefined network of interest (see methods). Finally we then repeated the analysis by modeling both *Distrust* and *ExpDistrust* (the latter as nuisance covariate). Under this setting we confirmed the interaction between source of the feedback (*Target vs. MPs*) and *Distrust* previously observed (x = 14, y = 20, z = 6, $t_{(43)}$ = 4.49, *p*

(uncorrected) <0.001, Cluster size = 308, *p* (whole-brain correction at the cluster level) = 0.002). Unfortunately, we did not confirm the simple negative linear relationship between the VS activity evoked by *Target* agreement and *Distrust*. We therefore extracted the data from the same coordinates reported in the main analysis and observed that, with the addition of the new covariate, the strength of the effect fell just below the threshold (x = 0, y = 10, z = 4, $t_{(43)}$ = 2.97, *p* (uncorrected) = 0.002, *p* (small volume correction) = 0.078). Please note, however, that this specific negative modulation can be considered redundant to our other findings (especially those in *Figure 6*). As such, it is not reported in the main results section, and its presence/absence does not alter for the main conclusions of this study.

Overall, the present follow-up analyses provide compelling evidence that the effects implicated by *Distrust* were not confounded by a broader assessment of the experimental set-up, but reflect a true consideration about the authenticity of the facial expressions. None of the effects associated with *Distrust* were observable when modeling instead *ExpDistrust* as a predictor of interest. Furthermore, in all-but-one cases the effects of *Distrust* could be replicated when adding *ExpDistrust* as additional nuisance predictor.

**Appendix 1—table 1.** Demographic characteristics of the sample tested in the present study.

| | Experiment 1 | | | | Experiment 2 | |
|---|---|---|---|---|---|---|
| Group | Controls | YMS | IMS | EMS | Controls | EMS |
| *Size* | 30 | 30 | 30 | 30 | 26 | 26 |
| *Age (sd)* | 25.03 (4.75) | 21.37 (3.81) | 22.03 (1.40) | 24.25 (2.00) | 23.73 (4.11) | 24.15 (1.41) |
| *Males* | 12 (40%) | 6 (20%) | 10 (33.33%) | 10 (33.33%) | 13 (50%) | 9 (34.62%) |
| | Pilot 1 | | | | Pilot 2 | Pilot 3 |
| Group | Controls | YMS | IMS | EMS | Controls | Controls |
| *Size* | 29 | 56 | 32 | 38 | 24 | 23 |
| *Age (sd)* | 21.69 (3.52) | 20.11 (2.33) | 22.03 (1.43) | 24.34 (1.68) | 23.54 (4.12) | 22.26 (3.86) |
| *Males* | 4 (13.79%) | 16 (28.57%) | 6 (18.75%) | 15 (39.47%) | 7 (29.17%) | 11 (47.83%) |

**Appendix 1—table 2.** Group differences in Empathic and personality traits, and in scores of pain sensitivity and coping.

Scores are described in terms of mean and standard deviation. Italicized values on grey background refer to significant group differences, as measured through one-way Analysis of Variance (Exp. 1, SPQ P: $F_{(3,116)}$ = 5.73, p=0.001; Exp. 2, BF C: $F_{(1,48)}$ = 8.80, p=0.005; all other variables: Exp. 1, $F_{(3,116)} \leq$ 2.35, p≥0.076, Exp. 2, $F_{(1,48)} \leq$ 2.96, p≥0.092). IRI: Interpersonal Reactivity Index; PT: Perspective Taking; EC: Empathic Concern; PD: Personal Distress; FS: Fantasy; SPQ: Situational Pain Questionnaire; P: Receiver operating characteristic curve probability; PCS: Pain Catastrophizing Scale; BF: Big Five Inventory; E: extroversion; A: agreeableness; C: conscientiousness; N: neuroticism; O: openness.

| | Experiment 1 | | | | Experiment 2 | |
|---|---|---|---|---|---|---|
| Group | Controls | YMS | IMS | EMS | Controls | EMS |
| IRI PT | 20.47 (3.87) | 20.83 (4.50) | 18.87 (4.12) | 20.37 (3.62) | 19.76 (9.93) | 21.76 (4.28) |
| IRI EC | 20.40 (4.91) | 22.70 (3.46) | 21.30 (2.83) | 21.90 (2.92) | 20.80 (4.16) | 20.24 (4.24) |
| IRI PD | 12.77 (5.86) | 10.30 (4.40) | 10.37 (3.76) | 10.53 (5.46) | 12.32 (6.19) | 10.48 (4.25) |
| IRI FS | 18.03 (5.60) | 20.47 (4.76) | 17.87 (5.55) | 18.73 (4.69) | 17.96 (7.18) | 17.96 (5.86) |
| SPQ P | *0.89 (0.08)* | *0.94 (0.05)* | *0.94 (0.04)* | *0.94 (0.05)* | 0.92 (0.05) | 0.94 (0.04) |
| PCS | 20.27 (10.04) | 21.70 (5.45) | 22.93 (7.72) | 17.97 (6.52) | 20.56 (9.92) | 19.84 (5.70) |
| BF E | 29.00 (5.13) | 28.53 (6.39) | 28.53 (6.85) | 27.33 (6.82) | 24.76 (6.22) | 27.00 (6.79) |
| BF A | 38.57 (5.95) | 38.87 (6.58) | 38.87 (5.59) | 41.77 (3.75) | 40.16 (4.04) | 41.96 (3.92) |
| BF C | 33.47 (4.97) | 35.03 (5.63) | 33.23 (6.00) | 33.10 (6.80) | *31.08 (5.09)* | *35.20 (4.72)* |
| BF N | 23.13 (6.25) | 22.10 (6.41) | 22.10 (6.50) | 21.23 (7.14) | 21.68 (7.19) | 19.68 (6.13) |
| BF O | 38.87 (5.88) | 37.53 (6.60) | 37.17 (6.45) | 37.27 (6.99) | 35.64 (6.36) | 36.84 (6.69) |

**Appendix 1—table 3.** Regions implicated when observing painful facial expressions.

As default regions are displayed if surviving correction for multiple comparisons for the whole brain at the cluster level. Entries in italic refer to regions surviving only small volume correction for brain structures implicated in previous meta-analyses on the same paradigm (*Kogler et al., 2020*). Coordinates (in standard MNI space) refer to maximally activated foci as indicated by the highest t value within an area of activation: x = distance (mm) to the right (+) or the left (−) of the midsagittal line; y = distance anterior (+) or posterior (−) to the vertical plane through the anterior commissure (AC); z = distance above (+) or below (−) the inter-commissural line. L and R refer to the left and right hemisphere, whereas M refers to medial structures.

| | SIDE | X | Y | Z | T | Cluster size |
|---|---|---|---|---|---|---|
| **Parametric modulation of Ratings (positive effect)** | | | | | | |
| Anterior Insula (AI) | L | −38 | 20 | 0 | 3.71[†] | 2 |
| Middle Cingulate Cortex | M | -4 | -4 | 26 | 4.80 | 187[*] |
| Amygdala (Amy) | L | −20 | -8 | −10 | 4.74 | 489[***] |
| Periaqueductal Grey | M | 6 | −26 | −12 | 3.78 | |
| Fusiform Gyrus | R | 34 | −72 | −10 | 4.09 | 652[***] |
| Superior Occipital Gyrus | R | 32 | −80 | 18 | 4.51 | |
| Middle Temporal Gyrus | R | 44 | −58 | 0 | 4.81 | |
| **Parametric modulation of Ratings: Controls > EMS** | | | | | | |
| *Anterior Insula (AI)* | L | −36 | 12 | -8 | 3.72[†] | 3 |
| dorsal Ant. Cingulate Cortex (dACC) | M | -6 | −38 | 8 | 4.30 | 203[*] |
| Caudate | L | -8 | 12 | -6 | 4.59 | 227[*] |
| Caudate | R | 8 | 14 | -6 | 4.12 | 219[*] |
| Basal Forebrain | R | 18 | 8 | −16 | 4.51 | |

[***]p < 0.001; [**]p<0.01; [*]p<0.05 family-wise corrected for the whole brain.
[†]p<0.05 small volume corrected for conjoint activation between self and others' pain from *Kogler et al., 2020*.

**Appendix 1—table 4.** Target.
Regions modulated linearly by the absolute distance between participants' ratings and the Target's Feedback position. All regions survived correction for multiple comparisons for the whole brain at the cluster level, or small volume correction for brain structures implicated in previous meta-analyses on the same paradigm (*Wu et al., 2016*).

| | SIDE | X | Y | Z | T | Cluster size |
|---|---|---|---|---|---|---|
| **Feedback discrepancy** | | | | | | |
| dorsal-post. Medial Prefrontal Cortex (dpMPFC) | M | 6 | 54 | 38 | 6.84 | 692[***] |
| Supplementary Motor Area | M | 8 | 16 | 68 | 6.01 | 792[***] |
| Ant. Insula/Inf. Front. Gyrus (AI/IFG) | R | 46 | 26 | -6 | 4.42 | 1437[***] |
| Temporal Pole (TP) | R | 48 | 14 | −28 | 7.08 | |
| Middle Frontal Gyrus (MFG) | R | 56 | 28 | 32 | 4.32 | 219[*] |
| Temporal Pole (TP) | L | −54 | 14 | −26 | 4.70 | 184[*] |
| Occipital Pole | L | −16 | −86 | -8 | 7.75 | 1327[***] |
| Occipital Pole | R | −14 | −88 | 2 | 6.19 | |

*Continued on next page*

*Appendix 1—table 4 continued*

| | SIDE | Coordinates | | | T | Cluster size |
| --- | --- | --- | --- | --- | --- | --- |
| | | X | Y | Z | | |
| **Feedback agreement** | | | | | | |
| Ventral Striatum (VS)/Putamen | L | -8 | 10 | -8 | 6.09 | 254** |
| Ventral Striatum (VS)/Putamen | R | 14 | 14 | -6 | 6.26 | 724*** |
| ventral Medial Prefrontal Cortex (vMPFC) | M | −12 | 38 | −12 | 4.64 | 373** |
| Middle Cingulate Gyrus | M | 6 | 0 | 36 | 4.17 | 184* |
| Superior Frontal Gyrus | L | −14 | 32 | 40 | 4.99 | 371** |
| Superior Parietal Cortex (SPC) | L | −26 | −52 | 62 | 5.51 | 7245*** |
| Postcentral Gyrus | L | −54 | −24 | 38 | 7.92 | |
| Middle Insula/Putamen | L | −24 | −10 | 6 | 4.47 | |
| Superior Parietal Cortex (SPC) | R | 26 | −52 | 68 | 6.42 | 6324*** |
| Postcentral Gyrus | R | 36 | −32 | 32 | 6.47 | |
| Middle Insula/Putamen | R | 26 | -8 | 6 | 4.80 | |
| Inferior Occipital Gyrus (IOG) | L | −30 | −90 | -8 | 6.96 | 4537*** |
| Inferior Temporal Gyrus | L | −46 | −64 | -8 | 5.13 | |
| Middle Temporal Gyrus | L | −56 | −56 | −10 | 5.22 | |
| Inferior Occipital Gyrus (IOG) | R | 34 | −88 | -8 | 8.01 | 3346*** |
| Inferior Temporal Gyrus | R | 54 | −50 | −20 | 4.20 | |
| Cerebellum | M | -2 | −56 | −24 | 5.13 | 705*** |
| **Feedback discrepancy: Controls > EMS** | | | | | | |
| Supramarginal Gyrus (SMG) | L | −64 | −48 | 30 | 4.73 | 547*** |
| **Feedback agreement: effect of Distrust (negative modulation)** | | | | | | |
| *Ventral Striatum (VS)* | M | 0 | 10 | 4 | 3.55† | 3 |

***p < 0.001; **p<0.01; *p<0.05 family-wise corrected for the whole brain.
†p<0.05 small volume corrected for the contrast 'agree >disagree' from **Wu et al., 2016**.

**Appendix 1—table 5.** MPs.
Regions modulated linearly by the absolute distance between participants' ratings and the MPs Feedback position.

| | SIDE | Coordinates | | | T | Cluster size |
| --- | --- | --- | --- | --- | --- | --- |
| | | X | Y | Z | | |
| **Feedback discrepancy** | | | | | | |
| dorsal-post. Medial Prefrontal Cortex (dpMPFC) | M | 8 | 38 | 38 | 4.32 | 272** |
| Caudate | M | 6 | 6 | 16 | 4.92 | 186* |
| Ant. Insula/Inf. Front. Gyrus (AI/IFG) | R | 42 | 26 | -8 | 5.08 | 471*** |
| Occipital Pole | L | −14 | −88 | -2 | 5.58 | 1002*** |
| Occipital Pole | R | 18 | −82 | -6 | 5.70 | |
| **Feedback agreement** | | | | | | |
| Superior Parietal Cortex (SPC) | R | 28 | −54 | 62 | 5.84 | 1531*** |
| Supramarginal Gyrus | R | 64 | −26 | 46 | 6.52 | |
| Superior Parietal Cortex (SPC) | L | −34 | −48 | 64 | 6.78 | 3023*** |
| Supramarginal Gyrus | L | −60 | −24 | 38 | 6.12 | |

*Continued on next page*

*Appendix 1—table 5 continued*

|  | SIDE | Coordinates | | | T | Cluster size |
|  |  | X | Y | Z |  |  |
| --- | --- | --- | --- | --- | --- | --- |
| Inferior Occipital Gyrus (IOG) | R | 47 | −70 | −10 | 8.16 | 2321*** |
| Inferior Temporal Gyrus | R | 52 | −46 | −22 | 4.78 |  |
| Inferior Occipital Gyrus (IOG) | L | −32 | −90 | −14 | 7.62 | 1542*** |
| Inferior Temporal Gyrus | L | −50 | −62 | -6 | 7.62 |  |
| **Feedback discrepancy: Controls > EMS** | | | | | | |
| Middle Cingulate Cortex (MCC) | M | 10 | −10 | 46 | 4.49 | 598*** |
| **Feedback agreement: effect of Distrust (positive modulation)** | | | | | | |
| Retrosplenial Cortex | R | 28 | −60 | 22 | 4.99 | 187* |

***p < 0.001; **p<0.01; *p<0.05 family-wise corrected for the whole brain.

**Appendix 1—table 6.** Target vs MPs.
Regions showing differential effects between the two feedbacks.

| **Feedback agreement: Target > MPs** | | | | | | |
| --- | --- | --- | --- | --- | --- | --- |
| Ventral Striatum/Putamen | R | 16 | 12 | -8 | 5.96 | 570*** |
| Middle Insula | R | 34 | 2 | 14 | 3.94 | 558*** |
| Caudate | R | 16 | -2 | 22 | 6.30 |  |
| Ventral Striatum/Putamen | L | −26 | 10 | -4 | 4.35 | 887*** |
| Middle Insula | L | −40 | -2 | 8 | 5.34 |  |
| Caudate | L | −20 | 2 | 22 | 5.03 |  |
| Thalamus | L | −18 | −20 | 18 | 6.08 |  |
| **Feedback agreement: (Target > MPs)*Distrust (positive modulation)** | | | | | | |
| Ventral Striatum | R | 14 | 20 | 6 | 4.94 | 227** |
| Supplementary Motor Area | M | -4 | 24 | 40 | 4.22 | 286** |

***p < 0.001; **p<0.01; family-wise corrected for the whole brain.
†p<0.05 small volume corrected for the contrast 'agree >disagree' from ***Wu et al., 2016***.

**Appendix 1—table 7.** Average non-grey matter nuisance covariate, defined through the coordinates whose grey matter probability is <0.02.
Table below represents the effects of a GLM similar to the one used in the main neuroimaging analysis but applied only to such estimate of nuisance signal with MarsBaR 0.44 toolbox (http://marsbar.sourceforge.net/). Group-level main effects were explored through one-sample t-tests on the GLM parameters. Groups differences were explored through two-sample t-tests, whereas linear effects of Distrust were analyzed through Spearman rank correlation. None of the effects were significant.

| Faces | No Grey Matter Signal (GM p < 0.02) |
| --- | --- |
| *Controls vs. EMS* | $t_{(50)} = -1.69$ |
| **Parametric Modulation (PM) of Pain Ratings** | |
| *Main effect* | $t_{(51)} = -0.78$ |
| *Controls vs. EMS* | $t_{(50)} = -0.99$ |
| **PM of Protagonist's Feedback Position** | |
| *Main effect* | $t_{(51)} = -1.90$ |
| *Controls vs. EMS* | $t_{(50)} = 0.94$ |

*Continued on next page*

*Appendix 1—table 7 continued*

| Faces | No Grey Matter Signal (GM p < 0.02) |
|---|---|
| *Distrust* | $\rho = -0.06$ |
| **PM of MP's Feedback Position** | |
| *Main effect* | $t_{(51)} = -1.18$ |
| *Controls vs. EMS* | $t_{(50)} = 1.97$ |
| *Distrust* | $\rho = -0.14$ |
| **PM of Protagonist vs. MP's Feedback Position** | |
| *Main effect* | $t_{(51)} = -0.92$ |
| *Controls vs. EMS* | $t_{(50)} = -0.24$ |
| *Distrust* | $\rho = 0.07$ |

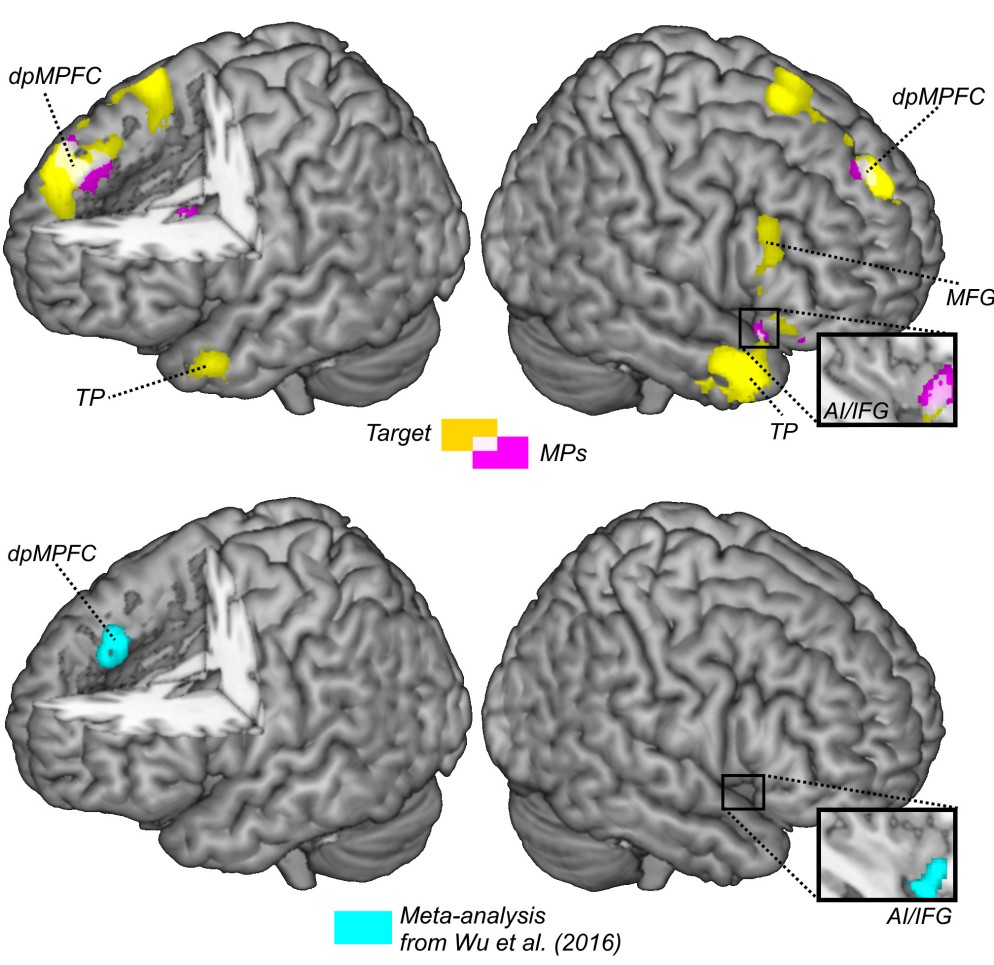

**Appendix 1—figure 1.** Surface renderings displaying regions implicated in the deviation from the social feedback. Yellow blobs refer to regions implicated in the deviation from the Target's feedback, whereas violet blobs refer to regions implicated in the MPs feedback. White blobs refer to regions implicated in both feedbacks. Finally cyan blobs describe regions implicated in disagreement from the social group in the meta-analysis from *Wu et al., 2016*. All effects are

*Appendix 1—figure 1 continued on next page*

*Appendix 1—figure 1 continued*

displayed under a height threshold corresponding to p<0.001, with each region surviving cluster-correction for multiple comparisons for the whole brain. dpMPFC: dorsal-posterior Medial Prefrontal Cortex; TP: Temporal Pole; MFG: Middle Frontal Gyrus; IFG: Inferior Frontal Gyrus; AI: Anterior Insula.

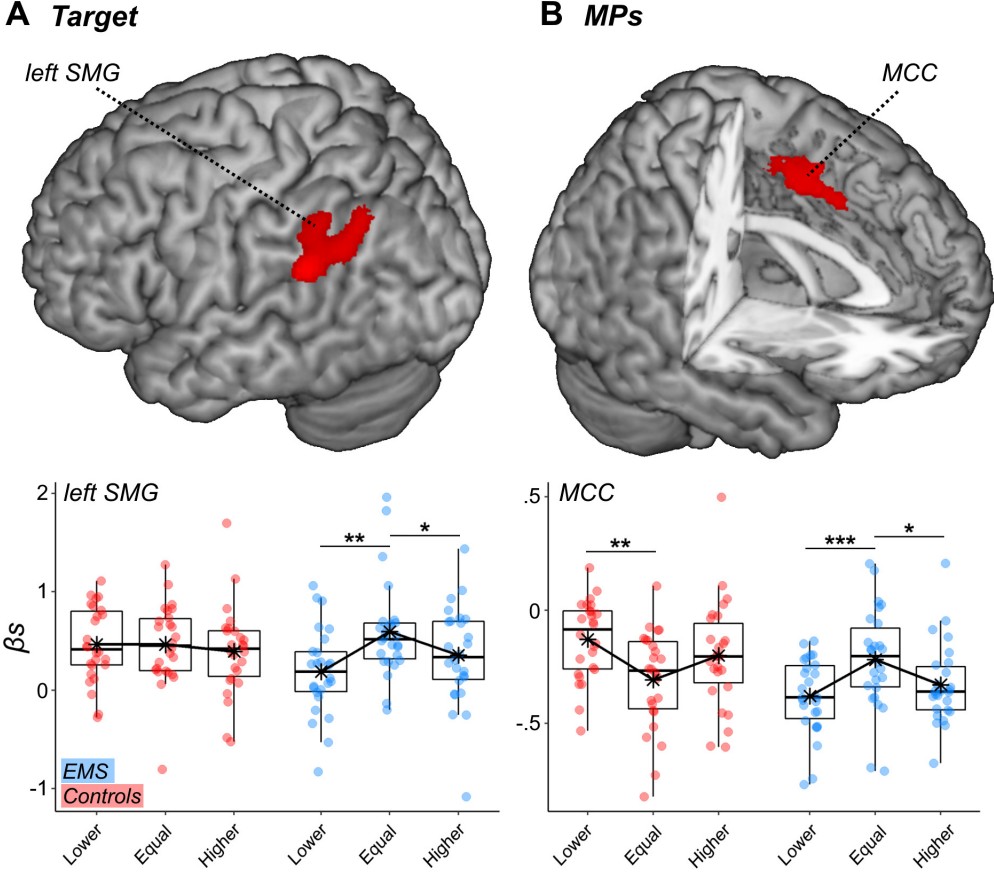

**Appendix 1—figure 2.** Surface renderings displaying regions showing group differences associated with processing (**A**) the Target's and (**B**) the MPs' feedback. Parameter estimates extracted from the outlined regions are displayed separately for Controls (red dots) and EMS (blue dots), and for their relative position (Lower, Equal, Higher) with respect to the initial ratings. All effects are displayed under a height threshold corresponding to p<0.001, with each region surviving cluster-correction for multiple comparisons for the whole brain. Note that feedbacks' position is displayed across three discrete categories to improve readability, although in the experiment it changed across a continuum (see methods). SMG: Supramarginal Gyrus; MCC: Middle Cingulate Cortex.

