## [Decision Letter]

**Acceptance summary:**

This study assessed differences between senior and junior medical students and individuals unaffiliated with the medical profession on pain assessment, finding that medical training is associated with reduced sensitivity to pain faces and decreased neural responses in the insula and cingulate cortex. This study identifies factors leading to pain underestimation and the effect of feedback regarding pain assessment in a medical setting, thus contributing important information about social and biological factors underlying interactions between doctors and patients.

**Decision letter after peer review:**

Thank you for submitting your article "Medical education and distrust modulate the response of insular-cingulate network and ventral striatum in pain diagnosis" for consideration by *eLife*. Your article has been reviewed by 3 peer reviewers, one of whom is a member of our Board of Reviewing Editors, and the evaluation has been overseen by Christian Büchel as the Senior Editor. The following individual involved in review of your submission has agreed to reveal their identity: Steven Anderson (Reviewer #2).

The reviewers have discussed the reviews with one another and the Reviewing Editor has drafted this decision to help you prepare a revised submission.

Summary:

This manuscript describes a study of senior medical students and younger colleagues assessing people's pain. The main findings are that medical training affects sensitivity to pain faces (indexed by lower ratings and decreased neural responses in the insula and cingulate cortices). These are interesting findings highlighting a phenomenon with real-world implications for medical practice. Previous studies on social influence have tested influences from one source, whereas here the authors test adjustments towards two (orthogonal) sources of influence (the person receiving pain/protagonist and a group of medical doctors [MPs]). The study assesses the effects of medical training by using a quasi-experimental design and reports that experienced medical students rate pain as lower and show reduced brain responses to others' pain. Distrust of patients/protagonists has an independent influence on pain reappraisals and brain responses. Overall, the reviewers agree that this is an important and highly interesting paper that aims to understand the neural mechanisms underlying the effect of medical training on patient pain underestimation. There are several notable strengths of this paper, namely the medical trainee sample at differing levels of experience, pilot study examining trainees' implicit attitudes towards medical practitioners (MPs), use of neuroimaging to understand underlying neural mechanisms of the effect of medical training on pain underestimation, and converging evidence in a large sample of participants over more than one experiment. Major concerns requiring additional analyses and revisions to the text should be addressed in the revision are listed below.

Essential revisions:

– There is an entire literature on racial disparities in pain underestimation (eg. physicians tend to underestimate pain in Black patients). This seems relevant and should be mentioned in the introduction as one aspect of the phenomenon under investigation. These biases are important to note as moderators of pain estimation.

– Relatedly, what was the racial composition of the participants and the faces used as stimuli in the experiment?

– It is not clear why medical students were the subjects of the study, rather than older more experienced physicians.

– The methods state that for neuroimaging analyses, "we also restricted our hypothesis based on regions previously implicated in the same paradigm". I think it would be important to report whole-brain results at the same threshold, without this masking, in order to more fully evaluate the strength of the reported effects.

– Given the applied nature of the research questions, I found the Introduction and Discussion to be somewhat lacking in the consideration of clinical relevance. I appreciated discussion of how the results were consistent or different from prior studies using social influence paradigms. However, I think a more clearly defined clinical relevance section is warranted given the applied research questions and medical trainee study sample. Within this section, the authors could also address the question of whether pain underestimation is necessarily bad, or whether physician emotion regulation serves some useful functions (e.g., to reduce cognitive load and burnout, Weilenmann et al., 2018, Front Psychiatry; Decety, J., Yang, C.-Y., and Cheng, Y. (2010), NeuroImage).

– I had a few issues with how Distrust was introduced, measured, and discussed in this paper. First, I think more justification in the Introduction for why practitioners might distrust patient pain report is warranted. Prior evidence that factors such as provider fear of patient opioid addiction and belief that women and racial/ethnic minorities exaggerate pain are relevant here (see review by Ghoshal et al., (2020) J Pain Res, work by Mende-Siedlecki et al., (2019), J Exp Psychol, and others). Several of these topics intersect with the broader issue of gender and racial/ethnic disparities in pain, which I realize is not the main focus of the present study. Nonetheless, this is a major applied area where provider pain underestimation is a known problem, and I think it could be acknowledged more in both the Introduction and Discussion to strengthen the clinical relevance of this paper.

– My other concern with Distrust in this paper is the discrepancy between how the concept was measured and how it is being described in the paper. The debriefing items used to define Distrust really seem to be measuring the realism or believability of the experimental paradigm, rather than the much more clinically relevant belief by the rater that the Protagonist is faking or exaggerating their pain. This discrepancy makes the conclusions drawn by the authors somewhat harder to understand.

– Furthermore, given that Distrust really only affected how feedback from MPs was taken into account, and not neural activity related to viewing Protagonist pain, is it possible that this measure is reflecting distrust or low-confidence in the rater's own judgments? This relates to an issue in the study design that does not appear to be addressed by the authors, which is the difference between the raters' level of medical training (notably not yet physicians) and the "average judgment of 20 emergency doctors." Is it possible that the raters were perceiving an authoritativeness in the 20 ER doctors due to both the number of doctors and the higher level of training of the doctors providing the feedback? Finally, more discussion of the fact that the distrust effect was regardless of medical training is warranted. Does this suggest that the factors that influence distrust of patient pain are more stable, such as implicit bias or beliefs about pain appropriateness? More consideration of the implications and limitations of the Distrust measure would help the reader contextualize the conclusions drawn.

– The authors controlled for participant age in models but did not address participant gender despite prior evidence of gender differences in empathic responding (Christov-Moore et al., 2014, Neurosci Biobehav Rev). Do the effects hold if also controlling for gender?

– One limitation that should be discussed more thoroughly is that the design remains quasi-experimental (and cross-sectional). Thus, despite the claims that medical training leads to reduced pain ratings, this is not entirely clear (and some kind of longitudinal 'training' or exposure study would be better suited to test this). First, there seem to be some group differences regarding sex. It is also not reported whether other important person-level variables may differ between medical students and controls (e.g., SES, IQ). Personality factors such as empathy, big five, PCS were assessed but not included in the Suppl. Table or elsewhere. Could any of those be related to the effects?

– Related to comment #1, it should be mentioned early on whether groups differed on other important variables, such as age, sex, education, etc.; Are controls also senior students, in another discipline than medicine? This applies to Experiment 1 (so far few details regarding potential group differences) as well as Experiment 2 (for which more details could be provided).

– The authors report having used global scaling (lines 666). This can dramatically alter the results of all statistical results and should be only done if there is no relationship between whole brain signal and task (which is very unlikely to ever be the case). I would recommend repeating the analysis without global scaling unless the authors can make a strong case (1) why it is needed here and (2) that there is no correlation between task and global brain activity.

---

## [Author Response]

Essential revisions:– There is an entire literature on racial disparities in pain underestimation (eg. physicians tend to underestimate pain in Black patients). This seems relevant and should be mentioned in the introduction as one aspect of the phenomenon under investigation. These biases are important to note as moderators of pain estimation.

We agree that racial bias is a very important aspect and is strongly linked to the mistreatment of pain in hospital settings. Although the present study does not address this issue specifically, we have now included dedicated paragraphs in the introduction and Discussion sections.

“As such, healthcare providers systematically underestimate patients’ pain (Davoudi et al., 2008; Duignan and Dunn, 2008; Kappesser et al., 2006; Puntillo et al., 2003; Teske et al., 1983), a phenomenon which emerges as early as during university (Xie et al., 2018), becomes more pronounced with long-lasting experience in the field (Choinière et al., 1990; Davoudi et al., 2008), and affects prevalently women (Greenwood et al., 2018) and ethnical minorities (Ghoshal et al., 2020; Kaseweter et al., 2012; Todd et al., 2000)” (Introduction, page 3, lines 10-13).

“This process […] exposes patients (especially women and individuals from ethnical minorities, Ghoshal et al., 2020; Greenwood et al., 2018; Kaseweter et al., 2012; Todd et al., 2000) to the risk of being unrelieved from their condition.” (Discussion, page 22, lines 7-12).

“Previous studies showed that confronting individuals from different social and ethnic group can impact profoundly representation of trust and associated activity in VS (Hughes et al., 2017; Stanley et al., 2012, 2011), but also sensitivity to their pain (Avenanti et al., 2010; Mende-Siedlecki et al., 2019) including the neural responses in AI and dACC (Cao et al., 2015; Hein et al., 2010; Xu et al., 2009). As such, the interplay between the reward system and AI-dACC network might offer a plausible model to explain why specific social/ethnic categories are less likely to be acknowledged for their pain and receive adequate treatment” (Discussion, page 26, lines 10-16).

– Relatedly, what was the racial composition of the participants and the faces used as stimuli in the experiment?

The video clips of facial expressions of pain were taken from Lamm and colleagues (2007) and include only Caucasian expressions. This information has been made explicit in the revised version of the manuscript.

“We used a database of 44 videos of painful facial expressions from Lamm et al., (2007) depicting Caucasian actors wearing headphones and simulating painful reactions to a high pitch noise (see Figure 1).” (Methods, page 28, lines 20-22).

Unfortunately, no data were collected about the racial composition of the participants. Therefore, we cannot test whether the ethnic concordance between the participants and the video-recorded people influenced the task performance. This has been acknowledgment in the portion of the Discussion section dedicated to the Limitations of the study.

“Yet, it is still possible that our results could have been influenced also by other factors untested in the present study. For instance, relevant missing information are the detailed educational level of Controls (including their year of enrollment at university) and participants’ ethnic group.” (Discussion, page 26, lines 4-7).

– It is not clear why medical students were the subjects of the study, rather than older more experienced physicians.

We are sorry if this issue was not described with sufficient clarity in the original version of the manuscript. Indeed, we agree that it would have been interesting to test professional physicians. This being said, several considerations influenced our choice and favored instead the selection of a student population. First, students have a major advantage in terms of feasibility, as they can be more easily recruited than adults (physicians, nurses, and also lay controls) who are often constrained under busy working schedules. Second, the recruitment of students at different years of enrolment allow for a clear-cut organization of homogeneous groups based on experience. This would have been less straightforward for professional physicians, whose experience cannot only be quantified in terms of years, but also vary at a qualitative level (as function of specialty, working department, part- vs. full-time engagement, etc.). In this perspective, any experiment recruiting physicians would have necessarily required restricting to specific subcategories, and relying possibly on smaller samples, not homogeneous in their different experience levels. Third, effects at the level of pain assessment and IAT D-Scores might very well manifest themselves already during university training, as suggested by some studies (Xie et al., 2018). As such, medical students represent the ideal target population to investigate the emergence of these effects.

Overall, although we acknowledge that scholarly and professional medical experience are not the same, we believe that medical students provide a relevant sample for our purposes, with benefits outweighing the limitations. In our revision we took care to make all these considerations explicit to the reader. Please check the revised versions of the Introduction and Results sections of the manuscript.

“As such, healthcare providers systematically underestimate patients’ pain (Davoudi et al., 2008; Duignan and Dunn, 2008; Kappesser et al., 2006; Puntillo et al., 2003; Teske et al., 1983), a phenomenon which emerges as early as during university (Xie et al., 2018), becomes more pronounced with long-lasting experience in the field (Choinière et al., 1990; Davoudi et al., 2008).” (Introduction, page 3, lines 5-10).

“This population was chosen due to its feasibility in terms of recruitment, and the straightforward quantification of individual experience (corresponding to the year of enrollment at university, regardless of future specializations, department, hierarchy, etc.). Furthermore, pain underestimation has been documented also in medical students (at least after 3 years, Xie et al., 2018), thus making this population suitable for our research question.” (Results, page 7, lines 4-9).

– The methods state that for neuroimaging analyses, "we also restricted our hypothesis based on regions previously implicated in the same paradigm". I think it would be important to report whole-brain results at the same threshold, without this masking, in order to more fully evaluate the strength of the reported effects.

We apologize if the original version of the manuscript was misleading for this specific point. Of course, as first step, we inspected results using always a common threshold, corresponding to FWE cluster-corrected p < 0.05. This is a well-established approach for inspecting whole-brain fMRI effects (Friston et al., 1993) which, under the current settings (SPM12 software, no image upscaling, 8 mm FWHM smoothing kernel, an underlying voxel-level threshold of p < 0.001 uncorrected), allows adequate protection from the α error (Flandin and Friston, 2019; Mueller et al., 2017). This was clarified in the revised version of the manuscript.

“As first step, we considered only those effects throughout the whole brain that exceeded p < 0.05, family-wise correction for multiple comparisons at the cluster level (Friston et al., 1993), with an underlying height threshold of p < 0.001, uncorrected.” (Methods, page 36, lines 9-11).

In three specific instances (two described in the main text), such threshold did not reveal significant effects in regions of theoretical interest for the contrast. For this reason, we followed established recommendations to apply small volume correction (SVC) within a specific a priori defined network (in our case based on meta-analytic ALE maps from previous studies using the same paradigm, see methods).

“As second step, we also restricted our hypothesis based on regions previously implicated in the same paradigms. Specifically, for the processing of painful expressions, we took the most recent meta-analysis on pain empathy from Kogler et al., (2020) […]. Instead for the processing of social feedback, we took the meta-analysis maps from Wu et al., (2016) […]. In both cases the meta-analytic activation maps (thresholded to survive FWE correction for the whole brain) were binarized and used for small volume correction in our study. Within these areas, we considered significant those effects associated with p < 0.05 FWE small volume correction at the voxel level.” (Methods, page 36, lines 11-24).

The reviewer fears that the use of multiple thresholds prevents the reader to fully evaluate the strength of reported effects. We apologize for this. As SVC was productively used only in specific instances (of course if hypothesized activations are visible already under whole brain correction there was no need to inspect the same network further), we decided to single out these cases so that any potential reader could clearly see them as exceptions. This was done in the text, figure legends and tables. In particular, in tables we italicized the results surviving only under SVC. In this perspective, non-italics entries correspond to the findings obtained when inspecting the whole brain under a common (cluster-corrected) threshold.

“…a positive linear relationship at the level of amygdala extending to the periaqueductal grey (PAG), and to the fusiform gyrus (Figure 3, green blobs). Furthermore, when applying small volume correction in regions previously implicated in paradigms for pain empathy (Kogler et al., 2020, see methods) the left anterior insula (AI) was also found (see Supplementary Table S3 for full details). […] We found stronger modulations in Controls as opposed to EMS in the dorsal portion of the anterior cingulate cortex (dACC). Under small volume correction, we also implicated the left ventral AI (Figure 3, red blobs)” (Results, pages 9-10).

“Effects are displayed under a height threshold corresponding to p < 0.001, with each region surviving cluster-correction for multiple comparisons for the whole brain, or associated with a peak surviving small volume correction for a mask of interest (this is the case of the two insular activations).” (Figure 3 legend, page 10, lines 15-18).

“Table S3. Regions implicated when observing painful facial expressions. As default regions are displayed if surviving correction for multiple comparisons for the whole brain at the cluster level. Entries in italic refer to regions surviving only small volume correction for brain structures implicated in previous meta-analyses on the same paradigm” (Supplements, page 56, lines 1-4).

– Given the applied nature of the research questions, I found the Introduction and Discussion to be somewhat lacking in the consideration of clinical relevance. I appreciated discussion of how the results were consistent or different from prior studies using social influence paradigms. However, I think a more clearly defined clinical relevance section is warranted given the applied research questions and medical trainee study sample. Within this section, the authors could also address the question of whether pain underestimation is necessarily bad, or whether physician emotion regulation serves some useful functions (e.g., to reduce cognitive load and burnout, Weilenmann et al., 2018, Front Psychiatry; Decety, J., Yang, C.-Y., and Cheng, Y. (2010), NeuroImage).

We thank the reviewer for this precious feedback. We revised the manuscript accordingly, by expanding on the clinical relevance of our findings in relevant theoretical sections of the manuscript. In first step, we included a section of the discussion suggesting:

“The negative effect played by medical scholarly and professional experience in the neural response to others’ pain has been often interpreted in terms of decreased empathic response, possibly promoted by enhanced regulatory abilities (Cheng et al., 2007; Decety et al., 2010). […] This process has the beneficial effect of shielding physicians from the emotional weight sharing others’ suffering (Gleichgerrcht and Decety, 2014; Vaes and Muratore, 2013; Weilenmann et al., 2018), but at the same time exposes patients (especially women and individuals from ethnical minorities, Ghoshal et al., 2020; Greenwood et al., 2018; Kaseweter et al., 2012; Todd et al., 2000) to the risk of being unrelieved from their condition.” (Discussion, pages 21-22).

At the second step, we also modified portions of the Introduction and Discussion section addressing specifically the notion of Distrust. Also in this case, we expanded on these sections by highlighting their clinical relevance. Please see, for more details, to our reply to the subsequent point.

– I had a few issues with how Distrust was introduced, measured, and discussed in this paper. First, I think more justification in the Introduction for why practitioners might distrust patient pain report is warranted. Prior evidence that factors such as provider fear of patient opioid addiction and belief that women and racial/ethnic minorities exaggerate pain are relevant here (see review by Ghoshal et al., (2020) J Pain Res, work by Mende-Siedlecki et al., (2019), J Exp Psychol, and others). Several of these topics intersect with the broader issue of gender and racial/ethnic disparities in pain, which I realize is not the main focus of the present study. Nonetheless, this is a major applied area where provider pain underestimation is a known problem, and I think it could be acknowledged more in both the Introduction and Discussion to strengthen the clinical relevance of this paper.

Following our reply to the reviewer’s previous comment, we do acknowledge that the paper will benefit from a more in depth introduction and discussion of the notion of Distrust in relation to clinically-relevant topic, such as opiophobia and pain underestimation ethnic minorities. Please see the revised theoretical section of the manuscript.

“In clinical settings, acknowledging high levels of pain often leads to the prescription of strong analgesics, which however have contraindications for patients’ health (Buckeridge et al., 2010; Butler et al., 2016; Makris et al., 2014). Concerns for such side-effects (e.g., opiophobia) contribute to inadequate pain treatment in medical settings (Bennett and Carr, 2002; Bertrand et al., 2021; Corradi-Dell’Acqua et al., 2019), as healthcare providers prioritize those cases in which pain is unequivocally established. In this perspective, one study showed that doctors and nurses tended to underestimate pain in larger extent when presented with cues that patients might have lied or exaggerate their ratings (Kappesser et al., 2006).” (Introduction, page 4, lines 8-16).

“Our findings are particularly relevant in clinical settings, where pain appraisal is the basis for the selection of subsequent therapeutic procedures, including the prescription of strong (and potentially dangerous) painkillers. In this context, the deontological need to relieve patient’s pain is often counterweighted by the equally relevant need to prevent future side-effects and complications, a conflict that is often resolved by each individual based on personal ability to cope with uncertainty and sensitivity to errors (Corradi-Dell’Acqua et al., 2019). Obviously, a key source of information for these decisions is the reliability of available pain cues (e.g., is the facial expression genuine?) which, if estimated below a given threshold, can relieve caregivers from any struggle by prioritizing other clinical considerations over the management of patients’ pain. As unfortunate drawback, estimated reliability might be vulnerable to biases, including those related to the social and ethical condition of the patient. Previous studies showed that confronting individuals from different social and ethnic group can impact profoundly representation of trust and associated activity in VS (Hughes et al., 2017; Stanley et al., 2012, 2011), but also sensitivity to their pain (Avenanti et al., 2010; Mende-Siedlecki et al., 2019) including the neural responses in AI and dACC (Cao et al., 2015; Hein et al., 2010; Xu et al., 2009). As such, the interplay between the reward system and AI-dACC network might offer a plausible model to explain why specific social/ethnic categories are less likely to be acknowledged for their pain and receive adequate treatment. Future studies will need to explore this.” (Discussion, pages 24-25).

– My other concern with Distrust in this paper is the discrepancy between how the concept was measured and how it is being described in the paper. The debriefing items used to define Distrust really seem to be measuring the realism or believability of the experimental paradigm, rather than the much more clinically relevant belief by the rater that the Protagonist is faking or exaggerating their pain. This discrepancy makes the conclusions drawn by the authors somewhat harder to understand.

We apologize if in the original manuscript the items investigating disgust were described in a simplistic (and potentially misleading) fashion. The precise (French) questions asked in the final debrief in the original language were the followingL

1. J'ai eu l'impression que les gens de la vidéo ont simulé le désagrément. This translates as “I had the impression that the people in the video were simulating unpleasantness”.

2. J'ai eu l'impression que la douleur observée était réelle. This translates as “I had the impression that the pain observed was real”.

As the reviewer can see, these question address specifically the realism of the facial expressions, rather of the experimental paradigm. This is chiefly the case of Item 1 does explicitly ask participants whether the people in the videos were faking pain. The correct formulation (in both French and English translations) is now reported in the revised version of the manuscript (Methods, pages 31-32).

This being said, we do acknowledge that participants’ debrief ratings might (at least in principle) be confounded by considerations about the experimental set-up. To shed more light on this potential confound, we exploited an additional item from the debrief questionnaire, which assessed specifically participants’ distrust about other parts of the paradigm than the video-clips.

3. J’ai eu l’impression que le jugement soit du protagoniste du vidéo soit du médecin n’était pas réel. This translates as “I had the impression that the judgment from either protagonist or doctor judgments were not real”.

Importantly, the scores of this new item (hereafter ExpDistrust) correlated with those of Distrust, albeit in Experiment 1 this was only marginally the case (Experiment 1: Spearman’s ρ = 0.15, p = 0.097; Experiment 2: ρ = 0.42, p = 0.003). This seems to confirm the suspicion of the reviewer that assessments about the expressions’ authenticity are partly confounded by overall considerations about the experimental set-up.

At this point we decided to assess whether the effects associated with Distrust from our first submissions reflected genuine considerations about the facial expressions, or underlie a broader assessment of the paradigm. As first step, we repeated all the relevant analyses by replacing Distrust with ExpDistrust. In principle, if the effects described in the first submission reflected broad considerations about the experimental set-up, the same effects should be observed under this new predictor. This was never the case.

Subsequently, we tested again effects of Distrust by adding ExpDistrust as additional nuisance covariance of interest. In all-but-one cases the effect of Distrust remained significant despite the inclusion of the new predictor, thus confirming that the effects observed were independent by general considerations about the paradigm. The only exception was the effect reported in Supplementary Table S4 (last line). This was an activation obtained only under small volume correction and that shifted just below threshold following the inclusion of ExpDistrust as covariate. As such, this specific was removed by the Results section (but discussed on supplements), with no consequence for the overall conclusions of the study.

Overall, we hope we convinced the reviewer that the effects related to Distrust reflect a true consideration about the authenticity of the facial expressions, and not a broader assessment of the experimental set-up. In the revised manuscript, we included a brief sentence about this issue (see below), reminding to a new supplementary result section where all analyses involving ExpDistrust are described and commented (pages 45-47). In addition, we also included in supplements a full list of all question employed in the debrief session (pages 42-43).

“Finally, we carried out control analyses to assess whether the effects of Distrust were confounded by an overall lack of reliance towards the experimental paradigm (rather than the facial expression specifically). For this purpose, we reanalyzed Distrust in combination of another item form the post-experimental debrief assessing individual thoughts about another part of the task (the feedbacks). Full details are provided in Supplementary Results, and confirm that participants’ responses in our task were influenced selectively by considerations towards the authenticity of the facial expressions”. (Results, page 17, lines 14-20).

“Importantly, such effect is not confounded with participants’ overall distrust towards the experimental set-up, but reflect specific considerations of the facial expressions implemented (see Supplementary Results).” (Discussion, page 25, lines 4-7).

– Furthermore, given that Distrust really only affected how feedback from MPs was taken into account, and not neural activity related to viewing Protagonist pain, is it possible that this measure is reflecting distrust or low-confidence in the rater's own judgments?

If by “rater” the reviewer refers to the participants themselves who are rating the video-clips during the task, then we can reassure the reviewer that our results are safe from this possibility. First of all, as quantifying others’ pain is far from trivial, it is entirely possible that part of our sample felt low confidence about their own abilities. It is also in principle possible that such confidence could have biased the distrust post-experimental ratings (although we have no way to establish this). However, even if this was the case, it would not account for our results. Low confidence in one’s own ratings would explain why participants follow feedbacks in general (i.e., opinion of others rather than oneself), but would not explain why participants follow in privileged fashion the opinion of MPs, as opposed to the person who is feeling pain directly. Our results make sense only if we acknowledge that our Distrust predictor measures low confidence towards the person in the video rather than oneself.

Instead, if by “rater” the reviewer refers to the person in the video whose rating is presented as feedback, then the answer might be less straightforward. Although we asked participants to rate their distrust towards the authenticity of the facial expressions, it is entirely possible that their judgment could have generalized also to the feedbacks (someone who fakes their own facial expressions might very well fake also self-reports) or to the Target in general (this person seems untrustworthy). One way to partially address this issue would be to repeat the same argument used in our previous answer. I.e., in the debrief session, we asked participants to rate the authenticity of the feedbacks. As the effects of Distrust from our study were independent from those associated with this item, they relate specifically to considerations about the facial expressions (see our answer to the previous point from the reviewers). Unfortunately, this argument is only partially effective, as in the debrief we asked to estimate of the reliability of both feedbacks combined, and not that of the Target specifically. This serves well the purpose of controlling for the reliance on the experimental manipulation (as we did in the previous point) but does not account for the reliance on Target’s rating specifically.

We should stress, at this point, that the reviewer well-founded argument is not a treat to the main conclusion of the study. Ultimately, in many clinical settings, trust is modulated by several factors beyond perceived symptom authenticity. In our answers to previous points (and in the Discussion section of the manuscript page 26, lines 8-17) we already acknowledged how estimates of the expression reliability could be vulnerable to biases, including (but not limited to) those related to social/ethnic group. These modulations of distrust, although not strictly related on the pain expressed by the patient, are nonetheless expected to influence the way in which patients’ feedbacks are taken into account for the selection of a treatment. Likewise, also in our study, it is possible that Distrust about the pain expression could be influenced by a general unreliance about the Target, and this could affect in turn the way in which feedbacks are processed and taken into account for subsequent evaluations. Both in our original and present submission, we considered Distrust as a subjective experience, and as such vulnerable to personal biases and influences. This however does not undermine the importance of studying how this factor influences the different facets of pain management.

– This relates to an issue in the study design that does not appear to be addressed by the authors, which is the difference between the raters' level of medical training (notably not yet physicians) and the "average judgment of 20 emergency doctors." Is it possible that the raters were perceiving an authoritativeness in the 20 ER doctors due to both the number of doctors and the higher level of training of the doctors providing the feedback? Finally, more discussion of the fact that the distrust effect was regardless of medical training is warranted. Does this suggest that the factors that influence distrust of patient pain are more stable, such as implicit bias or beliefs about pain appropriateness? More consideration of the implications and limitations of the Distrust measure would help the reader contextualize the conclusions drawn.

The welcome the reviewer’s point that the social modulation observed in this study is well interpretable in terms of “authoritativeness”. The original studies employing this paradigm refer to “conformity” (Klucharev et al., 2009; Wu et al., 2016), although this does not fit entirely our case. Indeed conformity effects are expected to operate in privileged fashion towards in-group members (Stallen et al., 2013), but in our study (1) the target population (medical students, not yet doctors) does not belong to the same social group than the MPs who provide the feedback, and (2) the factors modulating the reliance of the MPs’ feedback operate regardless of the group (thus also in controls). In the original version of the manuscript we used the term “Social Influence” to characterize effects that could occur regardless of the belonging group. However, the reviewer offers a plausible alternative. MPs are not followed because “peers”, but rather because they are figures of authority which are more relied upon when both medical students and controls distrust the Target. In the revised version of the manuscript we discuss briefly about this.

“Distrust operates independently from the grouping factor and from the personal dispositions towards the category of doctors (measured through the IAT). Hence, our effects might not reflect experience, belongingness or personal positive dispositions towards MPs, but possibly the fact that doctors are figures of authority which become much more salient when the facial expression cannot be relied upon.” (Discussion, page 25, lines 7-11).

As for the number, we chose to define MP cues in keeping with previous studies using the same paradigm, in which social cues are often described as the combined information from different individuals from one community (Klucharev et al., 2009; Koban and Wager, 2016; Shestakova et al., 2013). This serves the purpose of making the social cue at the same time reliable and anonymous (if the opinion was described arising from one doctor only, this would raise questions about its’ identity/expertise/etc.). This being said, the reviewer is in principle correct, as the authoritativeness could reflect the number people (20 MPs vs. 1 patient) rather than their background. This is now acknowledged as limitation.

“in designing our study we followed previous paradigms in which social feedbacks (in our case MPs) were described as the aggregate opinion of many individuals from a community (Klucharev et al., 2009; Koban and Wager, 2016). This however opens the question as to whether Distrust promotes reliance towards MPs feedback due to their professional background, their size (an opinion of 20 is more reliable than that of one) or a combination of both factors.” (Discussion, page 27, lines 8-13).

– The authors controlled for participant age in models but did not address participant gender despite prior evidence of gender differences in empathic responding (Christov-Moore et al., 2014, Neurosci Biobehav Rev). Do the effects hold if also controlling for gender?

In the revised version of the manuscript we included gender (as well as age) as additional predictor of no interest in all analyses. Implementing such change had no impact in the results and the overall conclusion of our study. Please see the revised version of the manuscript where the inclusion of nuisance predictors is explicitly mentioned.

“…we tested whether the mean pain rating prior to the presentation of any feedback changed as function of Group through an ANOVA with Gender and Age as nuisance control variables. The analysis confirmed an effect of Group (F(3,114) = 3.09; p=0.030). […] We also found an effect of Gender (F(1,114) = 5.58; p=0.020), reflecting more pronounced ratings in female, relative to male, subjects (Age: F(1,114) = 1.27; p = 0.271).” (Results, page 9, lines 4-11).

“This was achieved by testing effects significantly associated with the parametrical modulation of pain ratings (whilst controlling for Gender and Age).” (Results, page 9, lines 19-20).

“No significance was associated with Group, Gender or Age (F ≤ 2.57, p ≥.053)”. (Results, page 13, lines 18-19).

“In our analysis we took care to account for several potential confounding variables, such as gender, age or empathic traits.” (Discussion, pages 26-27).

“Single trial Reappraisal values were fed to a Linear Mixed Model (LMM) with the relative position of the Target and MPs feedbacks as continuous predictors. Furthermore, we also included Group, Gender and Age as between-subject predictors, each modeled as both main effects and in interaction with Target and MPs.” (Methods, page 31, lines 11-14)

“Parameters associated with conditions of interest […] were then fed in a second level, independent-samples t-test using random-effect analysis. Furthermore, inter-individual differences related to the IAT D-Score or to distrust estimates were modelled through linear regression. In all analyses, Gender and Age were always modeled as nuisance regressors.” (Methods, page 36, line 4-9).

“An analysis of variance (ANOVA) with the D-score as dependent variable and Group, Gender and Age as independent variables revealed only a main effect of Group (F(3,148) = 3.19; p = 0.025; all other effects: F < 0.51; p > 0.475).” (Supplementary Information, page 44, lines 17-19)

– One limitation that should be discussed more thoroughly is that the design remains quasi-experimental (and cross-sectional). Thus, despite the claims that medical training leads to reduced pain ratings, this is not entirely clear (and some kind of longitudinal 'training' or exposure study would be better suited to test this). First, there seem to be some group differences regarding sex. It is also not reported whether other important person-level variables may differ between medical students and controls (e.g., SES, IQ). Personality factors such as empathy, big five, PCS were assessed but not included in the Suppl. Table or elsewhere. Could any of those be related to the effects?

We agree with the reviewer and apologize for not having stressed this limitation in the previous version of the manuscript. This was rectified this in the new “limitations” subsection of the Results:

“The present study investigates the role of medical experience by recruiting independent groups at different years of university. As such, our study shares the weaknesses of cross-sectional investigations (Wang and Cheng, 2020), as the role of experience was not tested longitudinally in the same population. In particular, some of our effects might be influenced by individual traits and features which are more frequently observed in medical students (especially those who reach the end of their tenure) as opposed to lay controls.” (Discussion, pages 26, lines 19-24)

The reviewer wonders whether group differences might be confounded by other variables than medical scholarly experience. We now acknowledge that this could be the case (at least in principle), and that our study controlled for this possibility only through a limited set of control measures. In particular, consistently with our reply to other points raised by the reviewers, we now included in all analyses age, gender and empathic traits as additional nuisance predictors, with almost no consequence for the results. Furthermore, we also included a new table (Supplementary Table S2) showing the scores for all questionnaires reported. For almost all of them, no group differences were found. As exceptions, the scores from the Situational Pain Questionnaire, suggest that Controls from Experiment 1 (but not Experiment 2) are less sensitive than the other groups to their own pain in imagined context; furthermore Controls from Experiment 2 (but not Experiment 1) have lower scores of conscientiousness (from Big Five Inventory) than medical students. We believe that these differences do not represent a serious threat to our results. This information has been included in the revised version of the manuscript.

“In our analysis we took care to account for several potential confounding variables, such as gender, age or empathic traits. Other measures (e.g., personality traits) appeared fairly stable across the different groups (with only few differences, not systematic across experiments; see Supplementary Table S2), and therefore are unlikely to have confounded our results. Yet, it is still possible that our results could have been influenced also by other factors untested in the present study.” (Discussion, pages 26-27).

“Table S2. Group differences in Empathic and personality traits, and in scores of pain sensitivity and coping. Scores are described in terms of mean and standard deviation. Italicized values on grey background refer to significant group differences, as measured through one-way Analysis of Variance […] SPQ: Situational Pain Questionnaire; P: Receiver operating characteristic curve probability; […] BF: Big Five Inventory; E: extroversion; A: agreeableness; C: conscientiousness; N: neuroticism; O: openness.” (Discussion, page 49, lines 1-9)

– Related to comment #1, it should be mentioned early on whether groups differed on other important variables, such as age, sex, education, etc.; Are controls also senior students, in another discipline than medicine? This applies to Experiment 1 (so far few details regarding potential group differences) as well as Experiment 2 (for which more details could be provided).

In the methods section we reported the principle with which we recruited lay controls.

“The first group comprehended lay individuals […] who were recruited among different faculties and professions, except those related to medicine, infirmary, dentistry and physiotherapy.” (Methods, page 28, lines 5-8)

In general we recruited participants though advertisements at University and word to mouth. Unfortunately, we do not know conclusively that all participants were university students. We did, however, asked them about their highest scholarly degree. In this perspective, Controls educational level ranged from those who finished high-school (Exp 1: 39%; Experiment 2: 48%), those who acquired a bachelor degree (Exp 1: 25%; Experiment 2: 36%), or a master degree (Exp 1: 36%; Experiment 2: 16%). However, a more systematic comparison with other groups is difficult as medical faculty in Geneva and Lausanne is organized into a unique 6-years cycle (with no difference between Bachelor and Master). For this reason, to account for difference in seniority between groups, we decided to focus only on age, for which controls are similar to the most experienced medical students. This being said, we now acknowledge the lack of detailed information about Controls’ educational level in the limitation section of the study:

“…relevant missing information are the detailed educational level of Controls (including their year of enrolment at university) and […].” (Discussion, page 27, lines 6-7)

Finally, the reviewer also expressed the wish that demographic information about the participants would be made available earlier in the text. We abided to this request and referenced a relevant table (Table S1) as early as the population of Experiment 1 is introduced in the manuscript.

“…we recruited 120 participants, organized as 30 Controls, 30 YMS, 30 IMS and 30 EMS. Supplementary Table S1 provides full details about demographic information of these participants, with controls displaying comparable age to EMS.” (Results, page 9, lines 1-3)

– The authors report having used global scaling (lines 666). This can dramatically alter the results of all statistical results and should be only done if there is no relationship between whole brain signal and task (which is very unlikely to ever be the case). I would recommend repeating the analysis without global scaling unless the authors can make a strong case (1) why it is needed here and (2) that there is no correlation between task and global brain activity.

We apologize if our analytical choices were not adequately justified in our first submission. Indeed, we originally reasoned that global variations of BOLD signal could reflect residual nuisance sources of variance independent of task manipulation, which are particularly pronounced in multiband sequences despite the correction for movement/physiological artefacts. The reviewer fears that the employment of global scaling might bias the results. Indeed, if the task effects are so widespread to influence the overall brain signal, then applying such correction might decrease the strength of the effects, or even produce artefactual deactivations (Aguirre et al., 1998; Desjardins et al., 2001; Junghöfer et al., 2005). In this perspective, it could be argued that also the outcome of the present study could be an artefactual effect of the global scaling.

To address this important point, we first assessed whether the global signal from our study is indeed influenced by the task. To do so, for each subject, we extracted the average time-course of the BOLD signal from the whole brain (as defined by the inclusive mask generated by the first-level models) and fed it to the general linear model similar to that used for the main fMRI analysis. This was achieved through the MarsBaR 0.44 toolbox (http://marsbar.sourceforge.net/; Brett et al., 2002) which allows to fit SPM-like models to the time-course of specific ROIs (in our case, the whole brain volume). Author response table 1 describes the results of group analyses. Indeed, as correctly suspected by the reviewer, the global signal was modulated by some (not all) of the contrasts of interest. Namely, in the analysis of the face-evoked signal, there was a significant group difference.

Furthermore, in the analysis of the feedback-evoked signal, there was a significant negative modulation of the protagonist’s feedback position, and such effect was significantly more pronounced than that of the MP’s feedback. Finally, the analysis of MP’s feedback position revealed also a significant group difference.

**Author response table 1. resptable1:** 

	Global Signal	No Grey Matter Signal (GM p < 0.02)
Faces		
Controls vs. EMS	t_(50)_ = -2.18*	t_(50)_ = -1.69
Parametric Modulation (PM) of Pain Ratings		
Main effect	t_(51)_ = 0.75	t_(51)_ = -0.78
Controls vs. EMS	t_(50)_ = 0.34	t_(50)_ = -0.99
PM of Protagonist’s Feedback Position		
Main effect	t_(51)_ = -2.74**	t_(51)_ = -1.90
Controls vs. EMS	t_(50)_ = 0.64	t_(50)_ = 0.94
Distrust	ρ = 0.04	ρ = -0.06
PM of MP’s Feedback Position		
Main effect	t_(51)_ = 0.13	t_(51)_ = -1.18
Controls vs. EMS	t_(50)_ = 2.04*	t_(50)_ = 1.97
Distrust	ρ = -0.20	ρ = -0.14
PM of Protagonist vs. MP’s Feedback Position		
Main effect	t_(51)_ = -2.32*	t_(51)_ = -0.92
Controls vs. EMS	t_(50)_ = -0.63	t_(50)_ = -0.24
Distrust	ρ = 0.21	ρ = 0.07
		**p < 0.01, *p < 0.05

Having established that global signal was indeed affected by the task, we sought for an alternative unbiased approach to account for nuisance sources of variance. In this perspective, we reasoned that task-independent global confounds should be expressed also in the signal outside the grey matter. We therefore modified the first-level analysis by removing the Global Scaling, and including instead the time course of the global signal outside the grey matter as a nuisance covariate.

Coordinates of interest were identified based on SPM tissue probability maps, by including those voxels with grey matter p < 0.02. The Author response image 1 provides a graphical representation of the identified regions, whereas Author response table 1 insures that the extracted average signal is not influenced by the task (at least not statistically). The reviewer could argue that the outlined regions represent only a sub-portion of the regions outside the grey matter, and that a more inclusive mask would be preferable. However, it should be underscored that any cut-off larger than p < 0.02 would have led to an average signal influenced statistically by the task, and therefore vulnerable to the same critique raised for the use of Global Scaling. In this perspective, it is reasonable to assume that BOLD signal from white matter might be partly confounded by neighboring grey matter structures, as a result of smoothing, suboptimal normalization or movement. Hence, only by focusing on coordinates reliably far away from the grey matter we could obtain an unbiased estimate of task-unrelated signal.

Finally, as last step, we inspected how the fMRI results were influenced by the changes in the first-level analysis. Overall, the results from the original version of the task were broadly unaffected by the modifications, with only two noticeable differences. On the one hand, the analysis of the face epochs was negatively impacted from the revision, with differential group effects no longer surviving correction for multiple comparisons for the whole brain, but only small volume correction for a network of interest (defined by previous ALE meta-analyses on similar paradigms Kogler et al., 2020). On the other hand, the analysis of the feedback epochs benefited from the revision, with some contrasts surviving now correction for multiple comparisons for the whole brain (and not only small volume correction).To sum up, we hope we convinced the reviewer that the results from the present study were neither artefactual to the scaling of global signal, nor idiosyncratic to the analytical pipeline employed in general. Despite some fluctuations in the strength of the effects, clearly the outcome of the study remains unchanged. For the purpose of this revision, we now replaced the original results with the new ones obtained when accounting only for task-independent non-grey matter signal. Please see the revised methods and Results sections for more details.

“To account for movement-related variance, physiological-related artifacts, and other sources of noise, we also included the 6 realignment parameters, an estimate of cardiac- and inspiration-induced changes in the BOLD signal based on PhysIO toolbox (Kasper et al., 2017), and the average non-grey matter signal, defined as the coordinates with grey matter tissue probability < 0.02. This is the largest non-grey matter mask whose average signal is not confounded by the task manipulation (see Supplementary Table S7), and as such represents an estimate of for global sources of noise that might can be accounted for without biasing the results (Aguirre et al., 1998; Desjardins et al., 2001; Junghöfer et al., 2005).” (Methods, page 35, lines 17-24)